# A missing source of aerosols in Antarctica – beyond long-range transport, phytoplankton, and photochemistry

M.R. Giordano[1], L.E. Kalnajs[2], A. Avery[1], J.D. Goetz[1], S.M. Davis[3,4], P.F. DeCarlo[1,5]

1 Department of Civil, Architectural, and Environmental Engineering, Drexel University, Philadelphia, Pennsylvania, USA

2 Laboratory for Atmospheric and Space Physics, University of Colorado at Boulder, Boulder, Colorado, USA.

3 Chemical Sciences Division, NOAA Earth System Research Laboratory, Boulder, Colorado, USA.

4 Cooperative Institute for Research in Environmental Sciences, University of Colorado at Boulder, Boulder, Colorado, USA.

5 Department of Chemistry, Drexel University, Philadelphia, Pennsylvania, USA

*Correspondence to*: P.F. DeCarlo (pfd33@drexel.edu)

**Abstract.** Understanding the sources and evolution of aerosols is crucial for constraining the impacts that aerosols have on a global scale. An unanswered question in atmospheric science is the source and evolution of the Antarctic aerosol population. Previous work over the continent has primarily utilized low temporal resolution aerosol filters to answer questions about the chemical composition of Antarctic aerosols. Bulk aerosol sampling has been useful in identifying seasonal cycles in the aerosol populations, especially in populations that have been attributed to Southern Ocean phytoplankton emissions. However, real-time, high resolution chemical composition data is necessary to identify the mechanisms and exact timing of changes in the Antarctic aerosol. The recent 2ODIAC (2-Season Ozone Depletion and Interaction with Aerosols Campaign) field campaign saw the first ever deployment of a real-time, high resolution aerosol mass spectrometer (SP-AMS or AMS) to the continent. Data obtained from the AMS, and a suite of other aerosol, gas-phase, and meteorological instruments, are presented here. In particular, this manuscript focuses on the aerosol population over coastal Antarctica and the evolution of that population in Austral spring. Results indicate that there exists a sulfate mode in Antarctica that is externally mixed with a mass mode vacuum aerodynamic diameter of 250 nm. Springtime increases in sulfate aerosol are observed and attributed to biogenic sources, in agreement with previous research identifying phytoplankton activity as the source of the aerosol. Furthermore, the total Antarctic aerosol population is shown to undergo three distinct phases during the winter to summer transition. The first phase is dominated by highly aged sulfate particles comprising the majority of the aerosol mass at low wind speed. The second phase, previously unidentified, is the generation of a sub-250nm aerosol population of unknown composition. The second phase appears as a transitional phase during the extended polar sunrise. The third phase is marked by an increased importance of biogenically-derived sulfate to the total aerosol population (photolysis of dimethyl sulfate and methanesulfonic acid [DMS and MSA]). The increased importance of MSA is identified both through the direct, real-time measurement of aerosol MSA and through the use of positive matrix factorization on the sulfur containing ions in the high-resolution mass spectral data. Given the importance of sub-250 nm particles, the aforementioned second phase suggests that early Austral spring is the season where new particle formation mechanisms are likely to have the largest contribution to the aerosol population in Antarctica.

# 1 Introduction

Although the present aerosol burden and processes are still relatively poorly understood, aerosol particles contained in ice cores obtained from the Antarctic ice shelves are used as proxies for many properties of the paleo-atmosphere. The abundance of mineral dust, sea salt aerosol (Fischer et al., 2007) and sulfur compounds (Legrand et al., 1988; Mulvaney et al., 1992) in Antarctic ice cores have been used to provide information on glaciation, sea level, cloudiness, and volcanic activity over the past millennia. Aerosols also play a major role in Earth's current climate due to their impact on the global radiative balance and cloud microphysics. Unfortunately, aerosols are still the least understood and constrained aspects of the climate system (Boucher et al., 2013). The uncertainty of aerosols' climate impacts arise from the fact that how an aerosol affects the radiative balance is a function of both an aerosol's chemical composition and physical properties (e.g. size, shape). Both the chemical and physical properties of aerosols are functions of emission sources, atmospheric processing, and lifetime in the atmosphere. Over recent decades, much work has been done to characterize aerosol emission sources and background aerosol across much of the globe (Boucher et al., 2013), but there are difficulties in in assessments of pre-industrial to present day forcing. Measurements in Antarctica, provide insight into one of the more pristine environments and can be useful in the understanding of preindustrial background aerosol (e.g. Hamilton et al., 2014). However, the ability to sample pristine aerosols is directly related to an areas inaccessibility. Because of the difficulty in performing science in Antarctica, the Antarctic aerosol mass and number population (particularly its sources and evolution) is still a subject of many open questions in atmospheric science. Improving our understanding of the processes that govern aerosol formation and evolution in Antarctica is important not only to our understanding of present day Antarctica, but also to understanding the broader climate history.

Besides a few research stations, bases, and minor tourism activities, anthropogenic emission sources of aerosols and trace-gases are rare in and around Antarctica (Shirsat and Graf, 2009). Much of the work produced over the years examining Antarctic aerosols has been done precisely because of the lack of direct anthropogenic atmospheric influences. Multiple investigations of the concentration, size distribution, spatial distribution, and composition of Antarctic aerosols have been performed over various parts of the continent (e.g., but not limited to, Shaw, 1979; Lechner et al., 1989; Harwey et al., 1991; Savoie et al., 1993; Hara et al., 1996; Minikin et al., 1998, Koponen et al., 2003). Low time-resolution bulk aerosol analysis (generally filter-integrated) dominates the chemical composition measurements over the continent (e.g. Prospero et al., 1991; Wagenbach, 1996; Minikin et al., 1998; Preunkurt et al., 2007) though real-time measurement techniques have been deployed in recent years (e.g. Belosi et al., 2012).

The results of aerosol measurements over Antarctica over the past decades have been generally consistent on two major points: that a persistent, low concentration, aerosol population exists over the entirety of the continent and that sulfate is a major component of that aerosol, especially in the Austral spring and summer (Shaw, 1979; Parungo et al., 1981; Wagenbach et al., 1988). On a global scale, anthropogenic sources of sulfur dominate the sulfate aerosol mass which has long been known to be a major component of aerosol induced climate forcing (Charlson et al., 1987; Charlson et al., 1990; Kulmala et al., 2002). However, in the Southern Hemisphere, and particularly in Antarctica, biogenic sources are thought to be the largest source of

sulfur to the atmosphere (Bates et al., 1992; Carslaw et al., 2010 and references therein). Hence, the question of the origin of the Antarctic aerosol sulfate arises. Globally, volcanic activity provides a major source of atmospheric sulfur so the presence of Mt. Erebus (78°S), the southernmost active volcano, should be noted. However, isotopic studies have shown that, in Antarctica, sulfate of volcanic origin is of minimal importance although infrequent, short-lived exceptions—due to eruptions—

have occurred (Patris et al., 2000). Descent of stratospheric sulfur into the lower atmosphere is another potential sulfur source but has also been shown to be a minor source of sulfate around coastal Antarctica (Legrand and Wagenbach, 1998). The two most likely remaining origins of sulfur are ocean sea-spray and non-sea-spray marine sources.

Marine sulfur appears in both primary and secondary aerosols. Primary marine aerosols, generally referred to as sea-spray aerosols, are produced through mechanical actions such as wave breaking and bubble bursting. Sea-spray aerosols are mostly

supermicron in size and production is a strong function of wind speed (Lewis and Schwartz, 2004). The supermicron sea-spray aerosol is dominated by sea salt, of which sulfate is 8% by weight in ocean water. In Antarctica, previous measurements have placed sea salt as 50-80% by mass of the (sub-10 μm) aerosol population depending on the time of year (Weller et al., 2008). Marine secondary aerosols, however, are driven by biological emissions of volatile organic compounds. Of these compounds, dimethyl sulfide (DMS) and its oxidation product methanesulfonic acid (MSA) are the most common and account

for 75% of the global biogeochemical sulfur cycle (Chasteen and Bentley, 2004). In Antarctica, enhancements in sulfate to sodium ratios (over the seawater ratio) in aerosols have been attributed to DMS and MSA. Off-line measurement and calculation of non-sea-spray (nss-) sulfate has been used to identify seasonal cycles of aerosol sulfate fractions over Antarctica (e.g. Prospero et al., 1991; Wagenbach, 1996; Minikin et al., 1998; Preunkurt et al., 2007). Spring and summertime enhancements in phytoplankton activity in the Southern Ocean do provide an excellent explanation for the spring sulfate

aerosol enhancements seen over Antarctica (Gibson et al., 1990; Arimoto et al., 2001; von Glasow and Crutzen, 2004; Preunkert et al., 2007; Read et al., 2008; Weller et al., 2011a). Unfortunately, non-sea-spray sulfate has generally been calculated using aerosol sodium concentrations. Because sodium is non-conservative in Antarctic aerosols due to mirabilite ($Na_2SO_4.10H_2O$) precipitation in sea ice microstructures, nss-sulfate calculations can be biased (Rankin et al., 2000; Rankin and Wolff, 2003). Direct measurements of gas phase DMS have been conducted over inland Antarctica (Concordia Station)

but without real-time aerosol measurements, it is impossible to determine the actual impact of marine biota on Antarctic sulfate (Preunkert et al., 2008).

Beyond sulfate and chemical composition information in general, knowledge of aerosol physical properties is necessary to constrain their climate effects. For example, quantifying the ability of aerosols to form cloud droplets is one of the key challenges in determining the overall climate effects of aerosols. This could be of particular importance in the Antarctic where

cloud formation may be limited by the low concentration of aerosols available to act as cloud condensation nuclei (CCN). Determining the CCN spectrum of a given aerosol population is possible once the size distribution, size-resolved composition, and mixing state of the aerosol population is known (Petters and Kreidenweis, 2007; Wang et al., 2010). Mixing state is one of the more difficult to measure properties of the aerosol spectrum. The extremes of mixing state are termed internal mixtures (all particles in a population, regardless of size, have identical chemical compositions) or external mixtures (there are multiple

particle populations with distinct and differing compositions) though most real aerosol populations are somewhere in between (Textor et al., 2006). The mixing state of an aerosol population affects cloud forming predictions when the aerosol population is comprised of components of significantly differing hygroscopicities (Wex et al., 2010). ==Because Antarctic aerosols seem to primarily be composed of sulfates and salts, the effect of the mixing state on cloud forming predictions may be minimized==

==over the continent itself but overestimated as continental air masses flow out over the Southern Ocean and gain organic components.==

The recent field campaign 2ODIAC—2 Season Ozone Depletion and Interaction with Aerosols Campaign—deployed a set of instruments that can begin to answer some of the outstanding questions about Antarctic aerosols; in particular, the questions surrounding high time resolution chemical speciation to determine the sources of the aerosol mass population. This is the first

manuscript from the campaign and focuses on the sources of Antarctic (coastal) aerosols and the physical properties of the sulfate aerosol mass population.

## 2 Methods

### 2.1 Field Site

The 2ODIAC campaign took place over two years with measurements occurring during the Austral spring/summer of 2014

(October-December) and the Austral winter/spring of 2015 (August-October). A field site was set up on the sea ice in McMurdo Sound approximately 20km from McMurdo Station, Antarctica. The 2014 field site was located at 77 41' 40" S, 166 11' 58" E while the 2015 site was 5 km away at 77 42' 58" S, 166 24' 30" E. The dynamic nature of sea ice prevented the collocation of the field sites over both field seasons. However, the 2015 site was chosen such that the wind fetch was similar to the fetch from the 2014 site. In 2014, the field site was located approximately 16km away from the sea ice edge as compared to

approximately 8km away in 2015.

In both field seasons, the field sites consisted of 2 structures ("fish huts"). One structure housed the atmospheric sampling instrumentation while the other housed a 5kW diesel generator. The generator hut was placed 75m to the southwest of the instrumentation hut because at both field sites the wind blew primarily from the northwest and southeast. The distance between and orientation of the huts ensured minimal self-sampling occurred over both field seasons. During low wind periods and

certain wind shifts, self-sampling and sampling of McMurdo did occur but these periods were identified and removed from the data via careful observation of $NO_x$ and AMS data. The thresholds applied in regards to self-contamination were if a) a rapid increase in NOx and associated decrease in ozone and b) the organic signal in the AMS at combustion-relevant $m/z$'s (e.g. $m/z$ 57 or 55) increased by greater than 20%.

### 2.2 Instrumentation

Atmospheric sampling instrumentation for both seasons consisted of a suite of aerosol- and gas-phase instruments. The gas-phase suite was composed of an ozone monitor ==(1 second time resolution;== Thermo Environmental model 49C) and a $NO_x$

monitor (1 second time resolution; Thermo Environmental model 42C). The aerosol suite consisted of aerosol sizing instruments, an aerosol concentration counter, aerosol composition measurements, and aerosol collectors. In 2014, aerosol sizing was carried out via a Scanning Electrical Mobility Spectrometer (~9-850 nm; 2-minute time resolution; Brechtel Man. Inc., SEMS) and Ultra-High Sensitivity Aerosol Spectrometer (~55- >1000 nm; 1 second time resolution; Droplet Meas. Tech.,

UHSAS). In 2015, aerosol sizing was carried out via a Scanning Mobility Particle Sizer (~10-420 nm; 2-minute time resolution; TSI, 3080/3081 SMPS and 3787 CPC) and an Optical Particle Counter (0.3-25 µm; 1 second time resolution; OPC - Lighthouse Remote 3104). In both years, total aerosol counts were measured with a water-based condensation particle counter (7 nm - >3 µm; TSI, 3783 Environmental Particle Counter - EPC) and aerosol composition was measured with a Soot Particle Aerosol Mass Spectrometer (Aerodyne Research Inc. Billerica, MA, SP-AMS, DeCarlo et al., 2006; Onasch et al. 2012). The SP-AMS is a combination of the Aerodyne High-Resolution Time-of-Flight aerosol mass spectrometer (HR-ToF-AMS) and a soot vaporizing laser (from Droplet Meas. Tech.). The soot vaporizing laser was not used in this study and the AMS operated on a 2-minute time resolution. The detection limit of sulfate in the AMS was calculated to be 10 and 12 ng m-3 for the 2014 and 2015 deployments, respectively (Jayne et al., 2000). The inlet for the aerosol sampling line was covered and heated to prevent sampling of wind blown snow and to prevent riming, respectively. At the flow conditions and geometry of the sampling inlet, transmission of <1µm particles to the AMS was >95%. Overall, the aerosol inlet system had a 50% transmission efficiency at 5µm and 0% transmission efficiency of particles >9µm. All transmission efficiency values are as calculated by a particle loss calculator using the specific geometry of the setup (von der Weiden et al., 2009). This work focuses on results from the aerosol size, concentration, and composition instruments primarily. Future work will discuss results from other parts of the instrumentation suite, including the particle filters and snow samples.

Meteorological data was recorded by a co-located weather station (Davis Vantage Pro2). The anemometer (rated 0-200 km h$^{-1}$), temperature and relative humidity probes (3˚C and 3% accuracy, respectively), and solar radiation sensors (spectral response 400-1100nm, 1 Wm$^{-2}$ resolution, ±5% accuracy at full scale) were mounted on a pole 50m NE of the sampling hut. It should be noted that, due to the sensitivity and accuracy of the radiometers, solar irradiance is presented in this manuscript as an approximate value. These values are useful to separate different temporal periods in the data but should not be taken as indicative of actinic flux values. Data from the sensors was wirelessly transmitted to the control unit located in the instrumentation hut.

During both field seasons, all of the instruments were routinely calibrated and, where applicable, verified against each other. For one hour each day, at different times every day, the inlet was switched to a HEPA filter for all of the particle instruments. This daily period allowed for a background signal to be calculated for the AMS as well as ensured no part of the inlet system was leaking (via monitoring of the particle counters/sizers). In depth calibrations and checks were conducted weekly. For the AMS, the weekly calibration included an ionization efficiency calibration using ammonium nitrate (atomized and size selected via the SEMS or SMPS at 300nm) and a PToF size calibration using PSL (60, 100, 300, 600 and 800 nm). Across both field seasons, the AMS calibrations changed by less than 10% while in the field. To check the particle counters against each other,

ammonium nitrate was size selected by the SEMS or SMPS and the output measured by the associated CPC's and EPC. Weekly zeros and spans were also conducted for the gas analyzers.

## 3 Results and Discussion

Figure 1 shows the records for wind speed, wind direction, number concentration from the EPC, and sulfate concentration from the AMS for both 2ODIAC field seasons. The wind speed, wind direction, and particle number concentration traces were collected every second but have been averaged to two minute records to match the AMS data recording rate. The average standard deviation of the The time series is displayed with the 2015 field season as the leftmost part of the x-axis to emphasize inter-seasonal transitions—from winter to spring (2015) and spring to summer (2014).

Figure 1a shows the wind direction (degrees) as a function of time and colored as a function of wind speed (m s$^{-1}$), for both seasons. The 2015 field season (winter-spring) was dominated by winds coming from the ESE with "high" wind speeds. Over 80% of the entire 2015 field season had wind coming from the ESE. Additionally, for greater than 60% of the 2015 field season had wind speeds recorded at over 8 m s$^{-1}$. By contrast, the 2014 field season (spring-summer) wind fetch had a more bimodal wind direction distribution and an opposite wind speed probability distribution as compared to 2015. In 2014, the wind direction distribution was 60/40% ESE/NW and was above 8 m s$^{-1}$ for only 20% of the season. These meteorological patterns and seasonal differences are not unusual for this region (Seefeldt et al., 2003).

Figure 1b shows the number concentration from the EPC over both field seasons. The figure shows the 2-minute average as well as a 1-hour average. With total condensation nuclei (CN) number concentrations ranging between 50-1000 cm$^{-3}$, 2ODIAC's observations are consistent with other coastal and interior Antarctic field measurements (e.g. Jaenicke et al., 1992; Gras, 1993; Koponen, et al., 2003; Belosi et al., 2012). Fig. 1b shows a major facet of Antarctic aerosols: that there is a steady-state aerosol concentration during calm and low-wind periods. The steady-state concentrations here are defined as the concentrations at which 99% of the recorded data is over. During the 2015 field season aerosol number concentrations were at a minimum of near 50 cm$^{-3}$ at the start of the campaign (early September). As the season progressed and the sun began to rise in the Austral spring, the background concentration of aerosols rose to 125 cm$^{-3}$. The 2014 field season experienced a similar minimum of approx. 75 cm$^{-3}$ but did not see the same rise in the background aerosol number population. Dotted lines at these values are included on Fig. 1b to aid the eye. A lack of combustion-derived organic aerosol signal in the AMS and the persistence of these particles independent of wind direction, suggests that the background aerosol is neither a result of local pollution (e.g. our own generators or vehicles) nor transported pollution from McMurdo.

Figure 1c shows the aerosol sulfate concentration measured by the AMS over the course of both field seasons. Aerosol sulfate positively tracks with the total aerosol counts both in the wind speed dependence and the trends in background concentrations. In 2015, total aerosol sulfate begins the season (in early September) at 20 ng m$^{-3}$ and rises to approx. 40 ng m$^{-3}$ as solar irradiance begins increasing at the end of September. In 2014, total aerosol sulfate increases from approx. 30 ng m$^{-3}$ (in October) to 60 ng m$^{-3}$ (in December) over the course of the campaign. The trend of an increasing sulfate aerosol burden in

springtime Antarctica is well documented. The values reported here are 2 to 3 times less than previous aerosol sulfate measurements in coastal Antarctica (Minikin et al., 1998; Rankin and Wolff, 2003). However, the previous measurements are filter-integrated assessments from Antarctic bases other than McMurdo (e.g. Halley, Neumayer, Dumont d'Urville) with differing size ranges and sampling techniques. Without co-incident measurements across the continent, it is difficult to determine if spatial or temporal factors are responsible for the differences in concentrations between literature values of aerosol sulfate and the presented values.

While aerosol sulfate is the main focus of this manuscript, it is not the only aerosol component and the relative amount of sulfate measured by the AMS should be contextualized. Over both field seasons, sulfate generally makes up more than 50% of the total mass of the traditionally reported non-refractory species (organics, sulfate, nitrate, and ammonium). Both the absolute amount and relative percentage of total non-refractory mass of sulfate is higher in 2014 than 2015. Ammonium, organics, and nitrate, in that order, make up the rest of the non-refractory species measured by the AMS. When adding measurements of refractory Na and Cl to the non-refractory species (as discussed in a forthcoming manuscript), sulfate is the third most abundant species at 5-30% of the total sub-micron aerosol mass.

## 3.1 Sulfate aerosols as an External Mixture

Taken together, the fact that neither total aerosol counts nor aerosol sulfate fall below a minimum value suggests that the background Antarctic aerosol number population may be primarily composed of sulfate. The relatively constant mass concentration of sulfate, independent of wind speed, wind direction and air mass origin, would indicate that this background aerosol is relatively temporally and geographically invariant. Other chemical species showed strong dependence on meteorological conditions in contrast to sulfate. If the background aerosol is primarily composed of sulfate species, then any enhancements in total aerosol counts would have to be a separate aerosol number population. An independent, externally mixed sulfate mode would therefore be expected in the AMS particle time-of-flight (PToF) mode. The PToF mode in the AMS is a mode of operation in which the size-resolved aerosol composition is determined by averaging many particles during the PToF operation time (2 minutes in this study). The PToF mode does not represent single particles but rather the size distribution of particles in which ions, such as sulfate ions, are present in. A large-scale, externally mixed sulfate mode would be expected to maintain a consistent size, during variations in wind speed and direction. In both 2014 and 2015, the (inorganic) sulfate species in the AMS exhibited such a wind-independent mass distribution. Figure 2 shows the sulfate vacuum aerodynamic diameter distributions from the AMS for the 2014 field season as a function of wind speed and direction. Regardless of where an air mass originated from, the sulfate aerosol as measured by the AMS exhibited a well-distributed mode centered at a vacuum aerodynamic diameter of approximately 250nm. This result is consistent with previous ocean based size measurements of sub-Antarctic MSA containing aerosol (250 and 370nm, from Zorn et al., 2008 and Schmale et al., 2013, respectively) as well as off-line filter integrated measurements in coastal Antarctica (200-350nm aerodynamic diameter impactor stage, Jourdain and Legrand, 2001). None of the other species measured in the AMS showed a well-defined size mode.

Similar to the AMS, the size distributions from the other aerosol sizing instruments were consistent across wind direction regimes as long as conditions conducive to blowing snow events (here defined as wind speeds greater than 8 ms⁻¹) are excluded. Figure 3 shows the average size distributions from the UHSAS, SEMS, and AMS from the 2014 field season when blowing snow events are excluded. For Fig. 3, all three measurements have been converted to a volume equivalent diameter to allow

intercomparisons of the three different methods of aerosol sizing by applying the following assumptions: 1) a density of 1.8gcm⁻³ for the sulfate aerosol population, 2) sphericity of all aerosols in the population, 3) equality of the optical diameter from the UHSAS and volume equivalent diameter (DeCarlo et al., 2004). As evidenced by the sizing instruments, the sulfate distribution is well encompassed by both the UHSAS and SEMS. For 2015 (not shown), the result is the same with the SMPS distribution compared to the AMS PToF. Without the size resolved sulfate distribution from the AMS, it would not be possible

to identify the externally mixed mode that is present.

The presence of an external mixture has implications both for estimating the bulk direct radiative forcing of Antarctic aerosol and in predicting CCN number concentrations ($N_{CCN}$) in the Antarctic troposphere. Many atmospheric radiation and global climate models assume external mixtures of aerosols (Koch et al., 2006; Bauer et al., 2007). Incorrectly assuming an externally mixed aerosol can result in incorrectly estimating radiative forcing by a factor of 3 or more (Bauer et al., 2007; Kim et al.,

2008). Recent improvements to radiative forcing models have focused on incorporating internally mixed aerosol populations (Boucher et al., 2013). The results presented here, although limited in seasonal coverage and duration of sampling, suggest that radiative forcing models for Antarctica should continue to treat the sulfate mass population as an external mixture. This work does support the assumptions of older estimates of radiative forcing for sulfate aerosols over Antarctica of approx. -0.1 Wm⁻² (Myhre et al., 1998). It should be noted that the appearance of sulfate primarily in the lower end of the size distributions

of Antarctic aerosol does not preclude the presence of sulfate in larger or smaller particles. The aerodynamic lens of the AMS inlet system precludes measurements of particles larger than 1 micron.

With regards to $N_{CCN}$, though size resolved composition has been used and is preferable, bulk composition data is often used to predict CCN number concentrations (Zhang et al., 2007). A simple volume weighted mixing rule such as the Zdanovskii−Stokes−Robinson relationship (ZSR, Stokes and Robinson, 1966) or the equivalent Kappa formulation (Petters

and Kreidenweis, 2007) is used to determine particle hygroscopicity. If non-size-resolved data from the AMS is used, the mixing rule implicitly assumes an internally mixed aerosol mass population and can bias calculated hygroscopicity high or low. In general, internal mixtures dominate aerosol populations as distance from emission source increase and the mixing rule is an appropriate assumption, regardless of the individual emission sources (e.g. Juranyi et al., 2010; Garbariene et al., 2012). However, when close to emission sources (especially anthropogenic sources), external mixtures are the generally observed

(Kander and Schultz, 2007; Swietlicki et al., 2008; Wex et al., 2010). The results here, however, are an externally mixed aerosol number population. The results presented here suggest that the Antarctic aerosol population may be a special case: despite a large distance between open water (presumably the major source of breaking waves near Antarctica) and the coast, an externally mixed aerosol population is persistent over the continent. Generally, there exists in the atmosphere enough mass to condense on pre-existing particles which reduces the impact of external mixtures downwind of the sources. Alternatively,

these results may suggest that open water is unnecessary for large ocean-atmosphere aerosol fluxes and that sea ice or other sources could be a dominant source of coastal Antarctic aerosol number population. The results presented here are limited by the fact that 2ODIAC occurred only on the sea-ice but the consistency of these results in context of the literature suggests that, at the least, they apply over a wider area of coastal Antarctica. It should be noted that previous studies in Antarctica that have measured $N_{CCN}$ have reported externally mixed CCN populations. Unfortunately, previous work has been conducted at Palmer Station or around the Antarctic Peninsula areas which have a strong and consistent marine influence (DeFelice, 1996; O'Dowd et al., 1997). This previous work has been unable to remove the marine influence from the continental air masses. An externally mixed sulfate mode, as this work demonstrates, appears to be a unique and ubiquitous feature of Antarctic aerosols. Calculating $N_{CCN}$ is not within the scope of this paper but will be presented in future publications on 2ODIAC.

### 3.2 Sulfate as a Predominant Component of the Background Antarctic Aerosol Number Population

If the externally mixed sulfate mode observed in the AMS is, in fact, the primary component of the background aerosol concentration, then some degree of closure between the EPC number concentration and the AMS PToF data should be possible. Again assuming particle sphericity and an average material density of 1.8 g cm$^{-3}$ for sulfate (an average value between sulfuric acid and ammonium sulfate), the total number of sulfate particles can be calculated from the size resolved sulfate mass distribution in the AMS PToF mode measured over the entirety of the campaign. This process assumes sphericity of particles takes the mass distribution measured in vacuum aerodynamic diameter space, and using the density value transforms it to volume vs volume equivalent diameter for that sulfate mode, which can then be converted into a number distribution and integrated to find the total number of particles from the measured sulfate mass distribution. Figure 4 shows the ratio of the total number of sulfate particles calculated from the measured AMS sulfate size distribution (average PToF size distribution scaled to the high resolution sulfate mass concentrations) to the total number of particles measured in the EPC. This number distribution calculation is assuming no counter-ion to the measured sulfate, i.e. the sulfate is present as sulfuric acid, which is consistent with previous Southern Ocean data (Zorn et al., 2008). As in Fig. 1a, the 2014 and 2015 field seasons have been combined and the data trace is colored by wind speed in ms$^{-1}$.

As discussed in the forthcoming Kalnajs et al. (2016) publication, higher wind speeds increase the non-sulfate aerosol counts. In discussing the background aerosol number population, any occurrence of blowing snow (implemented in this manuscript roughly as wind speeds above 8 ms$^{-1}$, as suggested by Li and Pomeroy (1997)) should therefore be ignored. For Fig. 4, this roughly translates to ignoring non-blue/green sections of the record. The total record is produced here for completeness but Fig. 4 also includes a smoothed trace (boxcar smoothing) that excludes high wind speed events (since high wind speeds and particle counts are strongly correlated, to be discussed in a forthcoming manuscript).

Sulfate modes can be described as three distinct phases as shown in Figure 4 (indicated by numbers and vertical lines). First, in the late Austral winter/early spring, sulfate comprises the majority of the total aerosol counts—often over 60% (1). Second, the percent of the total aerosol number population that is made up of sulfate then takes a sharp decline (2). Third, the percent sulfate gradually climbs again as the Austral spring transitions to summer (3). The spring/summer increase is primarily evident

in the 2014 data. While the fraction of particles that are primarily sulfate has a distinct cycle during these three phases, it should be emphasized that the sulfate mass is steadily increasing from winter to summer (Figure 1c).

Phases (1) and (3) are both in line with previous studies performed in and around Antarctica. Phase (1) is indicative of the low aerosol loadings and the persistent aerosol sulfate component (total, i.e. not nss-) seen multiple times over the continent in the winter (e.g. Gras et al., 1985; Wagenbach et al., 1988; Savoie et al., 1993). Phase (3) is likewise a measurement of the well-known springtime buildup of aerosol sulfate mass in and around Antarctica. Both direct measurements and model results have suggested that the sulfate buildup is due to phytoplankton activity in the Southern Ocean (e.g. Minikin et al., 1998, Khan et al., 2015). Phase (2), however, has not previously been identified as a major aspect of the annual evolution of the Antarctic aerosol number population. We note that this transitional season has only been observed in this dataset, and future observational datasets that measure during this period are important for understanding the consistency of this period.

Phase (2) is consistent with measuring newly formed particles that have been transported to our measurement location during a transitional period during the extended Antarctic sunrise. Throughout this manuscript, all instances of "new particle formation" refer to a regional, not local, growth of previously unobserved aerosol mass unless otherwise noted. The particles that are formed during this transitional period make up a significant fraction of the total aerosol number concentration at the very start of the Austral spring. The new aerosol population's contribution to the total Antarctic number population, however, decreases as biogenic emissions in the Southern Ocean increase.

The exact cause of Phase (2) is difficult to determine from the instrumentation deployed during 2ODIAC but potential explanations can be narrowed down. From Fig. 1 during Phase (2), both the total counts on the EPC and the sulfate mass in the AMS trend upward. However, total counts increases faster than the mass captured in the AMS. Three possible explanations therefore exist for the trends seen in Fig. 4 during phase (2):

1.) the particles are refractory and would not be measured by the AMS regardless of size,

2.) the particles are being counted in the AMS but are not sulfate (e.g. organics, nitrate), or

3.) there are particles measured by the EPC that are either not producing measureable size distribution signal or outside of the AMS measurement range on either the low (less than ~40 nm) or the high end (greater than 1 micron).

These three explanations for the transitional aerosol number population observed in Phase (2) must be examined before any conclusions on the source(s), impacts, or fate of the new number population can be drawn. A thorough discussion of these three possibilities is found in Appendix B but the main points are discussed here.

Explanation 1 (refractory particles) is unlikely to account for the transitional period aerosol due to the fact that the main source of refractory aerosols would be sea spray. Enhancements in sea spray would also cause enhancements in the mass and volume loadings due to the large contribution of sea spray to the super-micron particle distribution. None of these enhancements were observed in other instrumentation running during 2ODIAC. Explanation 2 (the transitional aerosol being non-sulfate) is unlikely since the ratio of sulfate to total non-refractory mass is steady when wind speed is accounted for (Figure A1). Explanation 3 (the transitional aerosol being a size not measureable by the AMS) is impossible to rule out. The reader is again referred to Appendix B for a complete discussion but the measurements from the sizing instruments (SMPS and SEMS) and

AMS suggest that the transitional aerosol is equal to or less than 250 nm in (vacuum aerodynamic) diameter. Unfortunately, given the instrumentation deployed, it is impossible to determine the composition of these very small particles (e.g. less than 100 nm) due in part to the low mass loadings inherent in Antarctica and the effects that low mass loadings have on AMS PToF data.

### 3.2.1 The potential sources of the transitional aerosol population

Phases (2) and (3) suggest that the photochemical processes affecting the Antarctic and Southern Ocean aerosol populations undergo at least two distinct phases. Unfortunately, determining the driving force behind either phase is not possible with the data available nor is there much data in the literature to explain the existence of Phases (2) and (3). For example, though the springtime MSA-derived aerosol increase has long been measured in Antarctica, data regarding what threshold of sunlight is necessary for the sulfate increase has been lacking. Data on Southern Ocean chlorophyll concentrations, light requirements for phytoplankton photosynthesis, and even solar irradiance measurements during Antarctic aerosol campaigns are sparse. During 2ODIAC, however, solar irradiance at the field site was measured for both field seasons though the values presented here should, again, be seen as approximate and indicative of a transitional period in the aerosol number population. The percentage of sulfate particles declines to a minimum of 20% when solar irradiance averaged between 100-200 Wm$^{-2}$ (i.e. most of the Austral spring after the sun had (partly) risen over McMurdo Sound). In the late spring/early summer transition, where daily peak solar irradiance averaged well over 300 Wm$^{-2}$, the percentage of sulfate aerosol begins to increase again.

Taken in conjunction with the sulfate data presented in Fig. 1 and the <40nm/total data in SI, Fig. 4 shows that during some transitionary period, measured during 2ODIAC as being between 100 to 300 Wm$^{-2}$, newly formed particles (and transport to the measurement site) may generate additional particles in the Antarctic number population. This Phase (2) to Phase (3) evolution of Antarctic aerosol has not previously been observed. Given the particle composition measurements, it is possible that significant contributions from non-sulfate (e.g. non-DMS related) aerosol formation mechanisms contribute to the Phase (2) aerosol. A likely explanation for the source of the new particles is the existence of some reservoir species that is photoactive during the early Austral spring. One likely reservoir species candidate for aerosol enhancement are halogens. Much work has recently been performed examining the importance of halogens and solar irradiance in coastal Antarctica. Gas-phase concentrations of IO and BrO have been shown to track well with solar irradiance levels at ~300 Wm$^{-2}$ (Saiz-Lopez et al., 2007). Solar photo-oxidation of frozen iodine containing solutions has been shown to increase gas-phase iodine concentrations as well (Kim et al., 2016). Halogens have long been linked to ozone depletion events (ODEs), and ODEs are linked to aerosol enhancements (e.g. Kalnajs et al., 2013). Unfortunately, the data are still unclear on if the spatially and temporally invariant aerosol increase is due to ODEs as they have been classically observed in the Polar regions. Another possible reservoir species in the Antarctic is mercury. Recent work has suggested that mercury can participate in new particle formation in the Antarctic atmosphere (Humphries et al., 2015). If reservoir species exist, then questions beyond their origin begin to arise. The mass of the reservoir, rates of source depletion and aerosol production, and aerosol formation mechanisms are all questions that would require additional measurements to answer.

Regardless of whether the non-sulfate aerosol is related to a mechanism such as ODEs or the existence of a reservoir species, two observations are clear: first, a (likely) sub-40nm mode of aerosols is somehow produced over coastal Antarctica in late winter/early spring and, second, the importance of the sub-40nm mode is reduced as sulfate mass in the larger aerosol modes increases in the late spring and early summer.

## 5  3.3 The Source of Antarctic Sulfate

Since the externally mixed sulfate mode observed in 2ODIAC makes up such a dominant fraction of the total aerosol number population, it is important to determine the source of the sulfate aerosol. As discussed, various other investigators have apportioned aerosol sulfate to either sea-spray or non-sea-spray sources usually through the use of sodium as a sea-spray marker. In addition to the problems of using sodium due to mirabalite fractionation, sodium is generally a refractory species in the AMS and not well captured at the standard operating conditions of the instrument. Sodium can be well measured by ion chromatography of filters but the record produced has long time integrations (24 hours in the case of 2ODIAC) and does not separate out wind and aerosol effects. Fortunately, the AMS is able to measure carbon-sulfur compounds in real-time, which have previously been tied to MSA concentrations in the marine boundary layer (Phinney et al., 2006; Zorn et al., 2008; Schmale et al., 2013).

MSA ($CH_3HSO_3$) is observed in the AMS as a marker ion at the mass to charge ratio ($m/z$) of 78.99 ($CH_3SO_2^+$). Unfortunately, due to fragmentation of the molecule on the vaporizer and in the ionization region of the AMS, the signal of $m/z$ 78.99 alone is not sufficient to determine the total MSA concentration (see MSA average mass spectrum, Figure S1 or Zorn et al., 2008). The total MSA signal can be reconstructed from total aerosol signal recorded in the field, however, by using the fragmentation pattern of MSA aerosolized directly into the AMS (Zorn et al., 2008). Figure 5 shows the reconstructed MSA concentration vs the total measured sulfate signal from the AMS, plotted for both field seasons. Also included on Fig. 5 is the data from two other field campaigns that measured Antarctic air masses with an AMS. The first is a reconstruction of total aerosol MSA (using an MSA fragmentation pattern) from a Southern Ocean cruise, adapted from Zorn et al. (2008). The second is an MSA-attributed factor reconstruction (via positive matrix factorization, see below) from a stationary deployment of an AMS to Bird Island, adapted from Schmale et al. (2013). The 2ODIAC data in Fig. 5 is colored as a function of the daily high solar irradiance measured at the field sites. As the sun reaches its approximate zenith, the MSA concentration in the AMS increases to almost 10% of the total sulfate signal in the AMS. During the late Austral winter and into the early Austral spring, the MSA fragment makes up less than 1% of the total sulfate. The absolute concentrations of MSA and sulfate of 2ODIAC are lower than those previously reported by both Zorn et al. (2008) and Schmale et al. (2013) though the ratios of MSA to sulfate are similar across the three studies. Both of the previous measurements took place in regions where the origin of particulate sulfate was dominated by open ocean source regions and took place in the austral summer and fall where more active DMS chemistry is likely to occur. Lower MSA and sulfate concentrations during 2ODIAC are therefore not surprising given the differences in season and location as compared to the previous studies. Higher MSA concentrations are likely to appear over McMurdo Sound as the sea ice recedes around the continent in the Austral summer (Jourdain and Legrand, 2001).

### 3.3.1 Positive Matrix Factorization

One advantage of the AMS over gas or ion chromatography in source apportionment is the ability to capture a full mass spectral signature of an aerosol mass population. By assuming the full spectrum is a combination of discreet components, the component spectra and mass loading of various sources can be determined using positive matrix factorization (PMF, Paatero and Tapper, 1994; Lanz et al., 2007). PMF application to AMS data has been described in detail elsewhere (Ulbrich et al., 2009; Ng et al., 2011; Zhang et al., 2011). For the results from 2ODIAC, PMF was applied to the high resolution data of the AMS. PMF analysis uses as input any number of mass spectra as a function of time, measured in the AMS, to pick out patterns in both temporal and mass spectral phases of the data. PMF determines "factors" that contribute to the time-series and mass spectral series by minimizing a "quality of fit" parameter (the sum of the residual of a given input not fit by the algorithm scaled by estimated error in the time and mass-spectral series). The PMF algorithm is run over multiple starting points in the rotations of the matrices and limited to given numbers of factors. $Q/Q_{exp}$ is used as a metric to determine how well the error model assumed in PMF is represented by the dataset. In general, the lowest values of Q/Qexp are the best solution. This metric is described in detail in Ulbrich et al. (2009). All of the measured sulfur containing $m/z$ ratios (excluding isotopes) were used as input to PMF to determine the sources of sulfur in Antarctic aerosols. PMF is generally performed on the organic species measured in the AMS, using only the sulfur compounds is a novel use of the method. PMF was run to explore solutions between 1 to 5 *factors* (sources of sulfur-containing aerosols) and numerous rotations of the data matrices (*fpeaks* between -8 to 8 at steps of 0.1). The major diagnostic plots and metrics are included in Supplemental Information.

Over all of the permutations of the input data, the final chosen solution resulted in 3 factors. It should be noted that the numbering of the factors is preserved from the PMF solution and is not indicative of any valuation on the factors themselves.

Figure 6 shows both the factor mass spectra and their time series reconstructions. The day where the daily high solar irradiance reached 300 $Wm^{-2}$ is also noted on the figure. Factor 3, the sulfate factor, is comprised of the traditional AMS sulfate peaks: $SO^+$ (47.97), $SO_2^+$ (63.962), $SO_3^+$ (79.96), $HSO_3^+$ (80.96), and $H_2SO_4^+$ (97.97). The MSA marker peak ($m/z$ 78.99) and $CH_2SO_2^+$ (77.98) are split between the Biogenic/MSA and Aged Biogenic Factors 1 and 2, respectively. The remainder of the carbon-sulfur complexed ions (e.g. $CHS^+$ (44.98), $CH_3S^+$ (47.00), $C_2H_3S^+$ (59.00), $CH_2SO_2^+$ (77.98), $CH_3SO_2^+$ (78.99), and $CH_4SO_3^+$ (95.99)) are predominantly found in Factor 2. Besides $CH_3SO_2^+$ and $CH_2SO_2^+$, Factor 1 is primarily composed of $SO^+$ (47.97), $HSO^+$ (48.97), $SO_2^+$ (63.96), $HSO_2^+$ (64.96), $SO_3^+$ (79.96), $HSO_3^+$ (80.96) and $H_2SO_4^+$ (97.97) (Fig. 6a). Factors 1 and 2 are differentiated in both their percent contributions of the CxHySzOp ions as well as the ratios of major sulfate peaks (e.g. $SO^+$:$SO_2^+$).

The naming of the factors, Sulfate, Biogenic/MSA, and aged biogenic, for Factors 3, 2, and 1, respectively, was chosen based on two main aspects of the resultant PMF spectra. The sulfate factor was named due to its lack of organosulfate complexes and the fact that it bears some resemblance to the mass spectra obtained by atomizing diluted sulfuric acid into the AMS. The Biogenic and Aged Biogenic factors were distinguished primarily by the predominance of the CHS complexes appearing in the Biogenic factor while not being completely absent in the Aged Biogenic factor. The aged biogenic factor is so named based

on the assumption that, in much the same way that aging organics eventually reduce to oxidized forms, organosulfur complexes may reduce to pure sulfate forms. If this assumption is not correct then the naming schema of the last factor may need to be adjusted in the future. Again, the three factor solution was chosen due to it minimizing the $Q/Q_{exp}$ of the PMF solution without creating obviously extraneous factors (see SI Fig.S4).

The Sulfate factor makes up the majority, 50-60%, of the total (reconstructed) mass over both field seasons (Fig. 6b). The Aged biogenic factor (factor 2) only has an appreciable contribution (10-20%) to the total non-refractory mass at the beginning of the Austral winter/spring field season in 2015, and ceases to contribute appreciable mass after September 23, when solar irradiance regularly and consistently breaks 100 Wm$^{-2}$. The aged biogenic factor has almost no influence on the Austral summer measurements. A possible source for the aged biogenic factor is long-range transport from areas of the Southern Ocean that

experience more sunlight earlier in the year than the Antarctic coast. The Biogenic/MSA factor, however, significantly contributes (40-50%) to the reconstructed mass during the Austral summer and, with the exception of October 2-5 (2015), has a steady contribution to the early Austral spring mass. Despite the minimal contribution of the aged biogenic factor, the three factor solution was chosen over the two factor solution for two reasons. The primary reason is the inadequacy of the 2 factor PMF solution with regard to MSA, which based on previous measurements in the Southern Ocean and the presence of a marker

ion (CH$_3$SO$_2^+$ at m/z 79) should make up some of the sulfur contribution. 2-factor PMF results either apportioned m/z 79 to 2 factors with 48:64 ratios that did not resemble any known substance (e.g. things tested in a lab setting included ammonium sulfate, pure MSA, diluted H$_2$SO$_4$, and southern ocean sea water) or apportioned majority of m/z 79 to a factor that was not temporally consistent with the CH$_3$SO$_2^+$ fragment in the dataset. The secondary reason for choosing the three factor solution is that three factors was consistently the number where diminishing returns in $Q/Q_{exp}$ began to occur. The attribution of the

MSA marker ion to the aged biogenic and biogenic/MSA factor indicates that both factors are likely representative of either MSA directly or of "biologically influenced" aerosols. Comparison to direct atomization of MSA into the AMS (see SI) suggests that the biogenic factor is made up of more than just MSA contributions since PMF did not find a "pure" MSA factor mass spectra for this dataset. The ratios of CH$_3$SO$_2^+$ to the major sulfate peaks (SO$^+$, SO$_2^+$, HSO$_3^+$, SO$_4^+$) in the two biogenic factors differ from pure MSA measured by the AMS in the laboratory.

Since using only the sulfur containing ions in PMF analysis is novel, it is difficult to compare these PMF results to previously published results. The closest related study is Schmale et al. (2013) which measured Antarctic/Southern Ocean air masses. In both the results presented here and in Schmale et al. (2013), the percent contribution of MSA to the total aerosol burden increases as sunlight (phytoplankton activity) increases over the Southern Ocean. Additionally, the MSA associated factor in that study is postulated to contribute significantly to the total sulfate signal, although it is not measured explicitly, which agrees

with the results here.

One major conclusion evident from the PMF solution is the relative lack of carbon-sulfur (C-S) complexes, other than CH$_3$SO$_2^+$, in the measured total sulfur (sub-micron, non-refractory) aerosol budget. The small amount of carbon-sulfur complexes present at the very end of the Austral winter are an interesting observation, and the low concentration of C-S complexes during the Austral spring and early summer (Aged biogenic, Fig. 6b) indicates that, either a) the aging and oxidation

of DMS/MSA may not occur over coastal Antarctica in the early summer or b) that the aging products of DMS/MSA complexes are present but not detectable by the AMS (e.g. see section 3.2.3).

As for MSA itself, the fact that an MSA containing fragment increases (in both absolute and relative terms) during the Austral spring and summer supports previous work that has suggested phytoplankton-derived sulfate is a major source of Antarctic sulfate aerosol. The PMF analysis supports the hypothesis that DMS and MSA are a primary component of sulfate aerosol over Antarctica in the Austral (late-) spring and (early-) summer. The PMF analysis also supports the observations of Fig. 4 with a marked increase in MSA-derived signal in the AMS after a threshold of solar irradiance is observed.

Combining Figs. 4, 5, and 6, 3 distinct phases of the aerosol number population over Antarctica become evident. The first is a highly aged sulfate population that makes up the majority of the total aerosol counts over the continent. The second phase is the increase of a new particle number population that occurs in the Austral spring. The third phase is the increasing importance of non-sea-spray sulfate (likely phytoplankton derived) with regards to the total aerosol number population. It remains to be seen if the trend reverses itself over the Austral Autumn as has been suggested in the measurement of MSA over Antarctica (Weller et al., 2011a and b).

## 4 Conclusions

The 2ODIAC campaign successfully deployed a suite of aerosol and gas-phase instruments, including the first ever deployment of an AMS to Antarctica, over two field seasons. The data from the AMS and particle counting instruments has shed new light on the aerosol mass and number population over the continent. First, a sulfate aerosol mode is shown to comprise the majority of the background aerosol population over 2 field seasons. Second, the background sulfate mass population is shown to be externally mixed with the distribution peaking at a vacuum aerodynamic diameter of 250nm. The 250nm mode was consistent throughout both seasons 2ODIAC was conducted over, regardless of local meteorology. Both of these conclusions together suggest that sulfate may be a relatively temporally and geographically invariant feature of the Antarctic aerosol population. Third, the 250nm sulfate population is shown to undergo three distinct phases with the Austral seasons. In the Austral winter, the 250nm mode dominates the total aerosol number population. In the late Austral winter/early spring a secondary aerosol mode of unknown composition comprises the majority of the aerosol number population. As Austral spring progresses into summer, the importance of the 250nm sulfate mode begins to recover to Austral winter levels of dominance of the aerosol number population. The Austral spring/summer buildup of sulfur is shown to be increasingly composed from MSA-derived aerosols, matching previous measurements over Antarctica. The unknown Austral spring aerosol number population is shown to be likely comprised of particles less than 250nm in size, possibly formed from halogen photochemistry or mercury catalyzed new particle formation over the Southern Ocean. This work further underscores the need to closely examine new particle formation and newly formed particles over Antarctica, and the Southern Ocean, in the early Austral spring.

**Acknowledgements**

The authors of this manuscript would like to thank the National Science Foundation for funding this work through grant numbers 1341628 and 1341492. Additionally, the authors extend many thanks to all of the support staff at McMurdo Station, especially Tony Buchanan; without their support none of this work would have been possible. The authors would also like to
5  thank Terry Deshler, Andrew Slater, and Anondo Mukherjee for their direct help with measurements in the field and Erin Frolli for her assistance in satellite image retrieval and creation. The authors would also like to thank and acknowledge Soeren Zorn and Julia Schmale for their permission in reproducing their data in Fig. 5. Any opinions, findings, and conclusions expressed in this material are those of the authors and do not necessarily reflect the views of NSF.

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

**Figures**

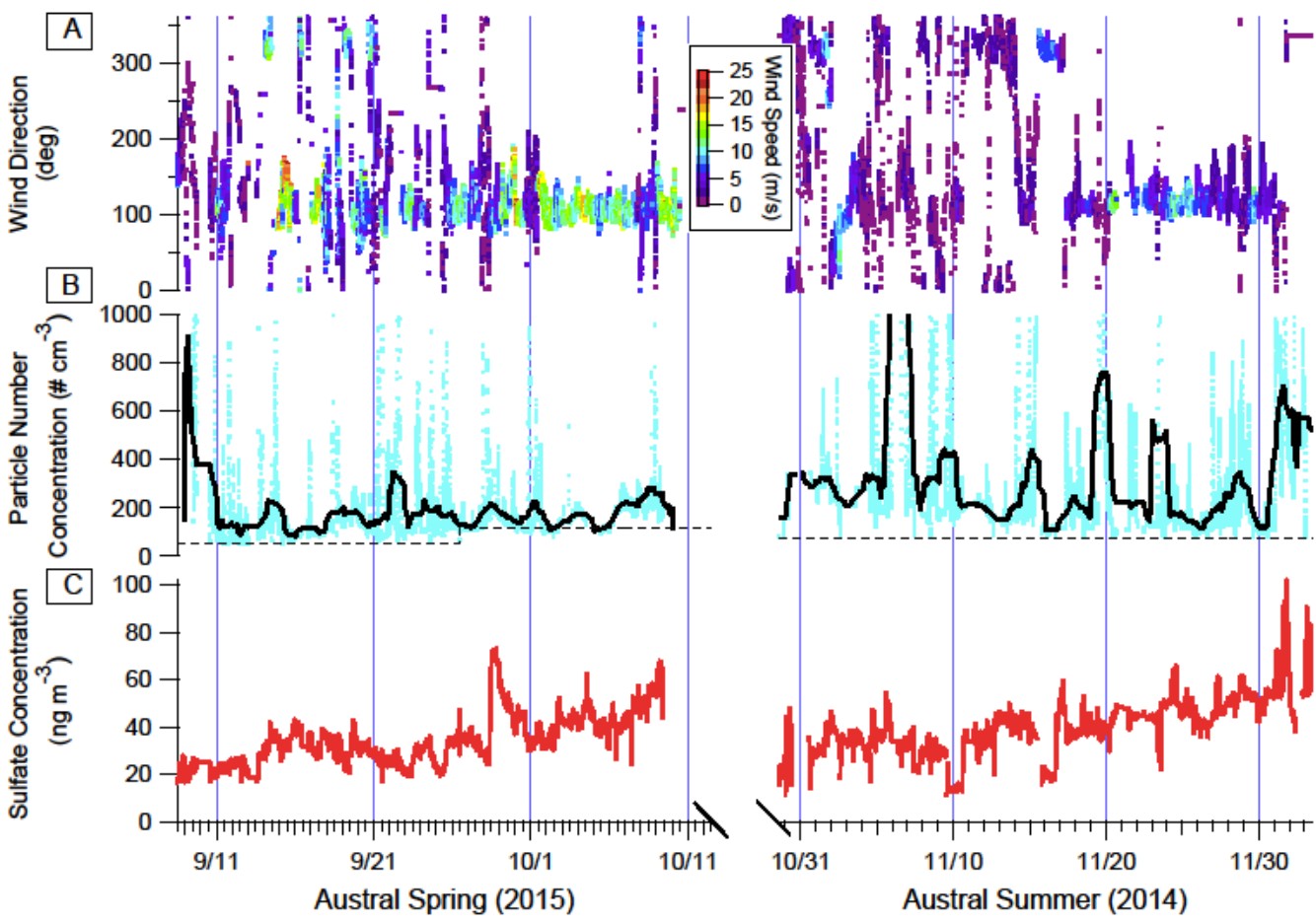

**Figure 1: For both the 2014 and 2015 field seasons, with 2015 leftmost: A) Wind direction record colored as a function of wind speed, displayed as a 2-minute average record (standard deviation 0.2 ms$^{-1}$ and 2 degrees) B) 2-minute (light blue; standard deviation of 11 and 18 # cm$^{-3}$ for 2015 and 2014, respectively) and 1-hour (black) records of particle number concentration from the EPC, C) 2-minute records of sulfate concentration from the aerosol mass spectrometer. Dotted lines on (B) indicate the minimums in particle number concentrations (99$^{th}$ percentile) measured over the field seasons.**

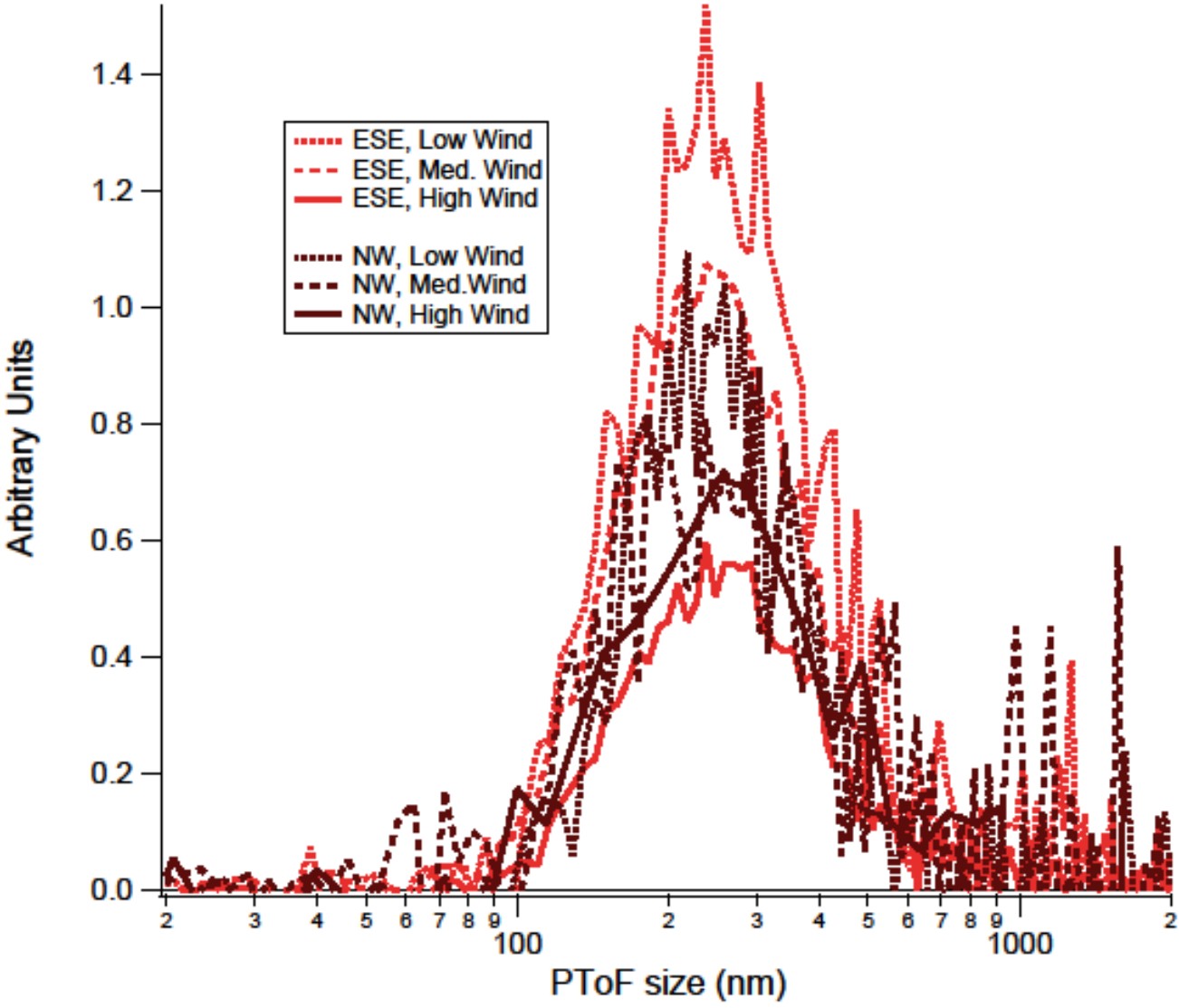

**Figure 2: AMS PToF results of AMS sulfate for 2014 as functions of wind speed (solid lines [> 8ms⁻¹] vs dotted lines [2-8 ms⁻¹ and <2 ms⁻¹]) and wind direction (black [Northwest] vs red [East Southeast]). The PToF mass has been arbitrarily scaled to enhance readability of the figure. The PToF results for 2015 are identical in both distribution shape and peak location for all wind regimes.**

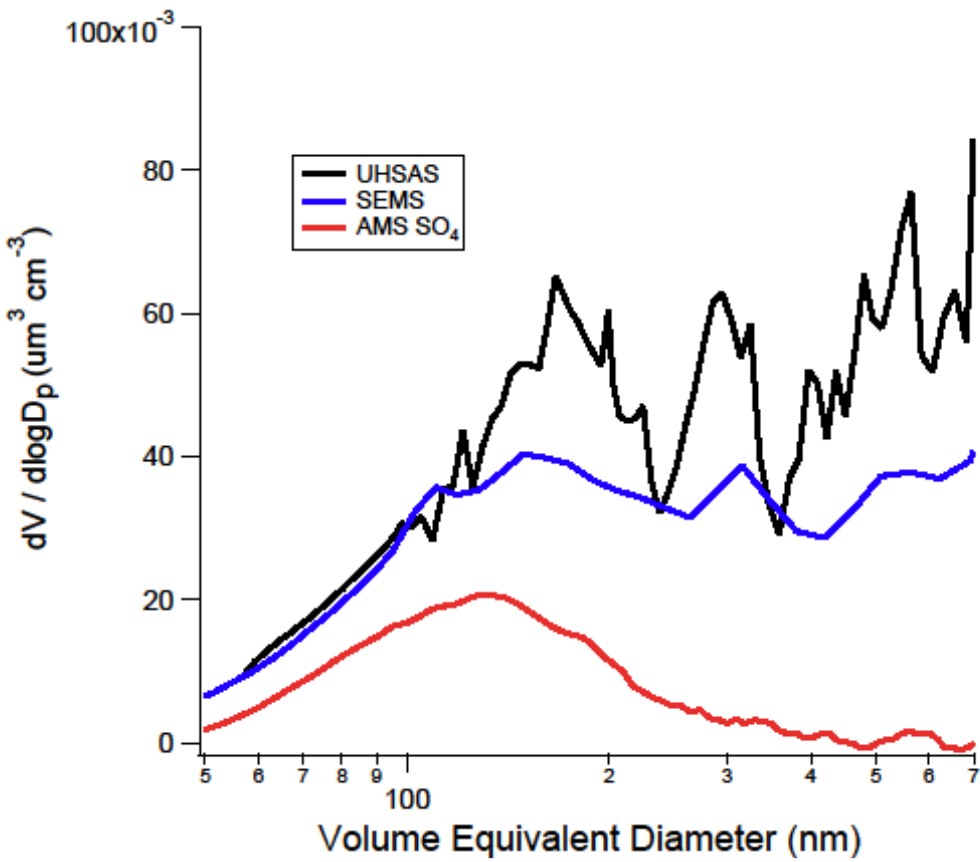

**Figure 3: Average dV/dlogDp as a function of volume equivalent diameter for low and medium wind speeds of 2014. Results of the Particle Time-of-Flight of the sulfate species from the AMS (red) are shown with the SEMS (black) and UHSAS (blue).**

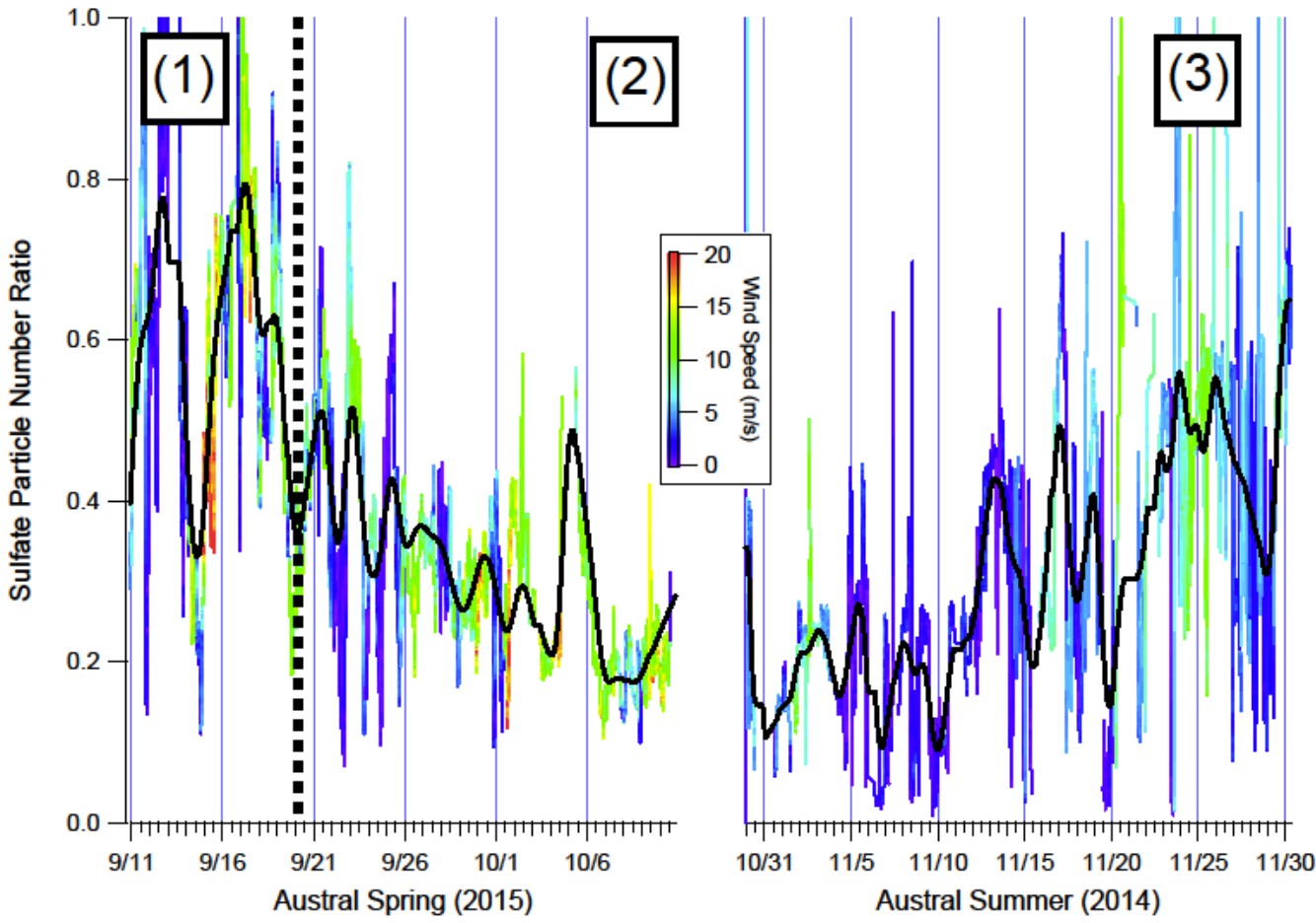

**Figure 4: Sulfate number ratio as calculated from the AMS PToF mode divided by the total number concentration from the EPC over both 2014 (right) and 2015 (left). Both field seasons data' are colored by recorded wind speeds. A smoothed trace of only wind speeds < 8m/s is overlaid in black. Rough timing for the phases discussed in section 3.2 are noted.**

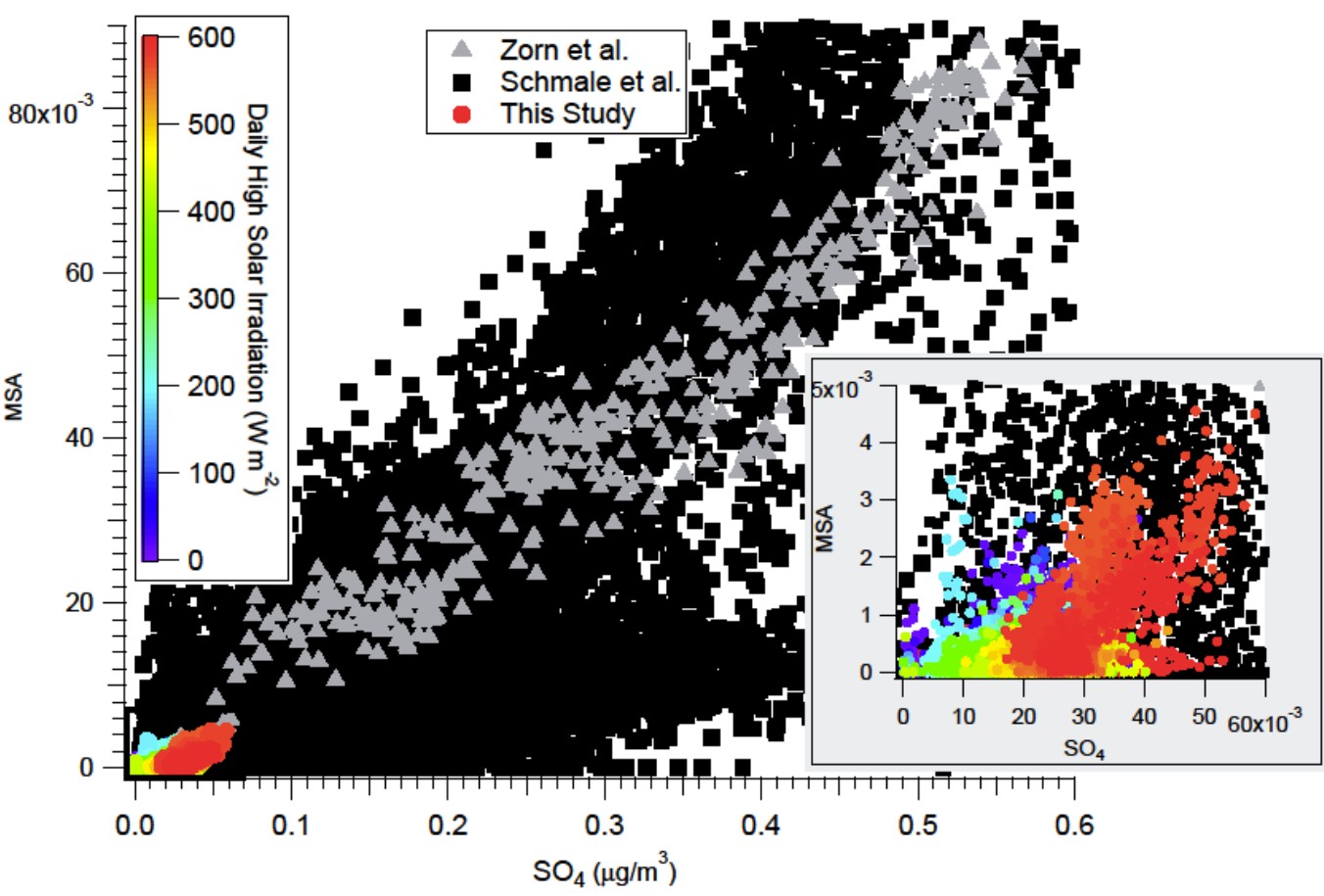

**Figure 5: MSA vs sulfate as measured by the AMS for both field seasons, colored by the daily high solar irradiance. 2ODIAC data is presented with results from Zorn et al. (2008) and Schmale et al. (2013) for context. Inset is a zoomed in area showing the 2ODIAC data in higher detail.**

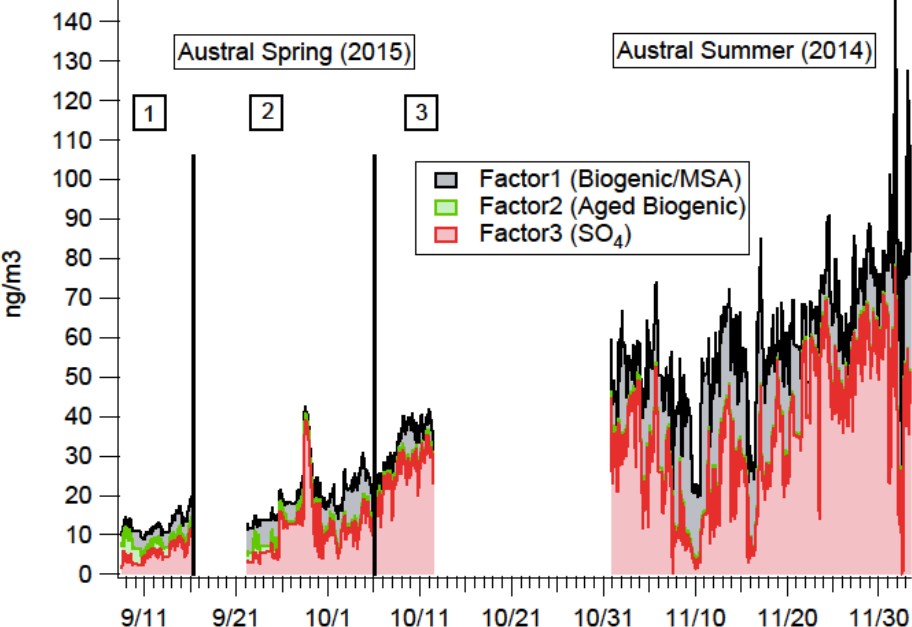

**Figure 6: PMF results for the sulfur containing species observed over both field seasons. A) Mass concentration reconstructions in ng/m3 for each factor and B) the mass spectral fingerprints for each of the 3 factors.**

**Appendix A:**

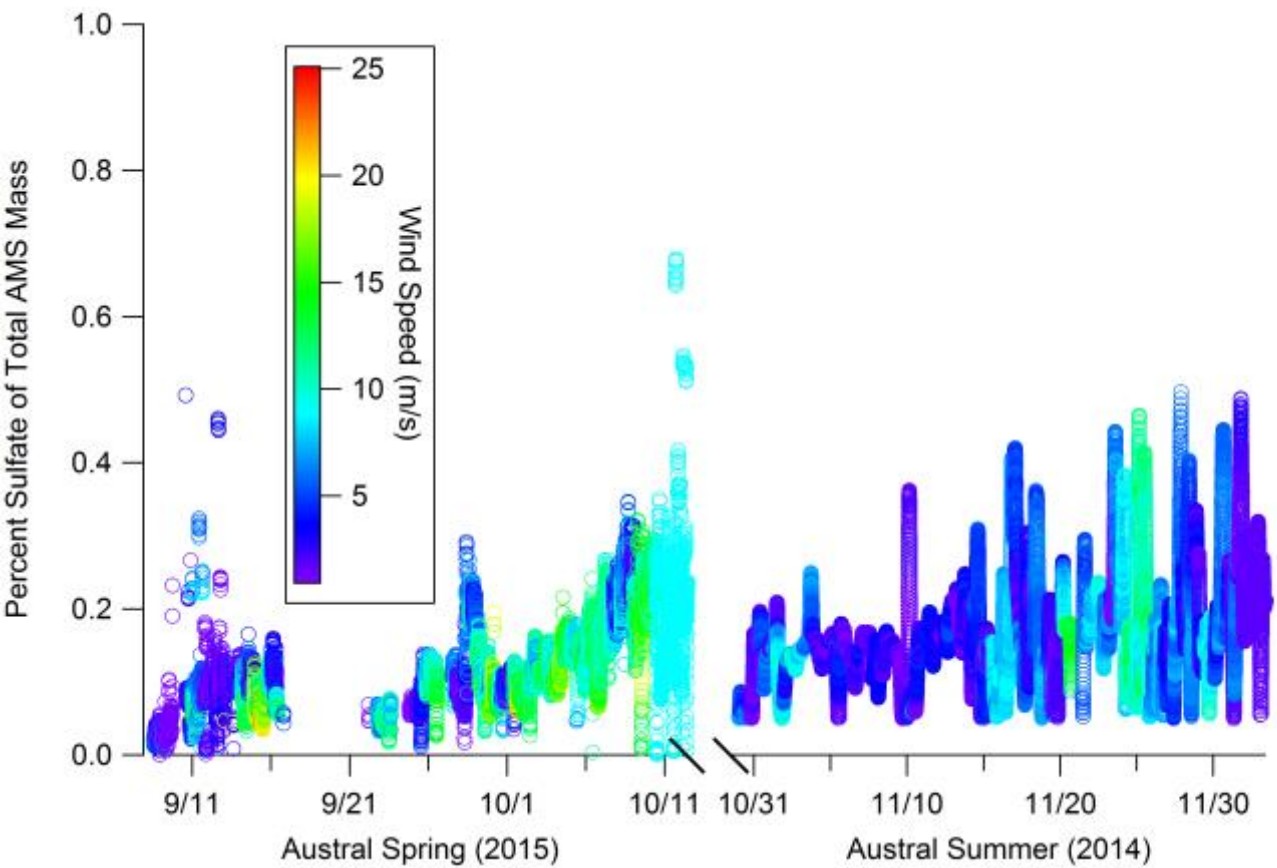

**Figure A1: Mass of sulfate as a percent of total non-refractory mass measured in the AMS, colored as a function of wind speed, for both field seasons.**

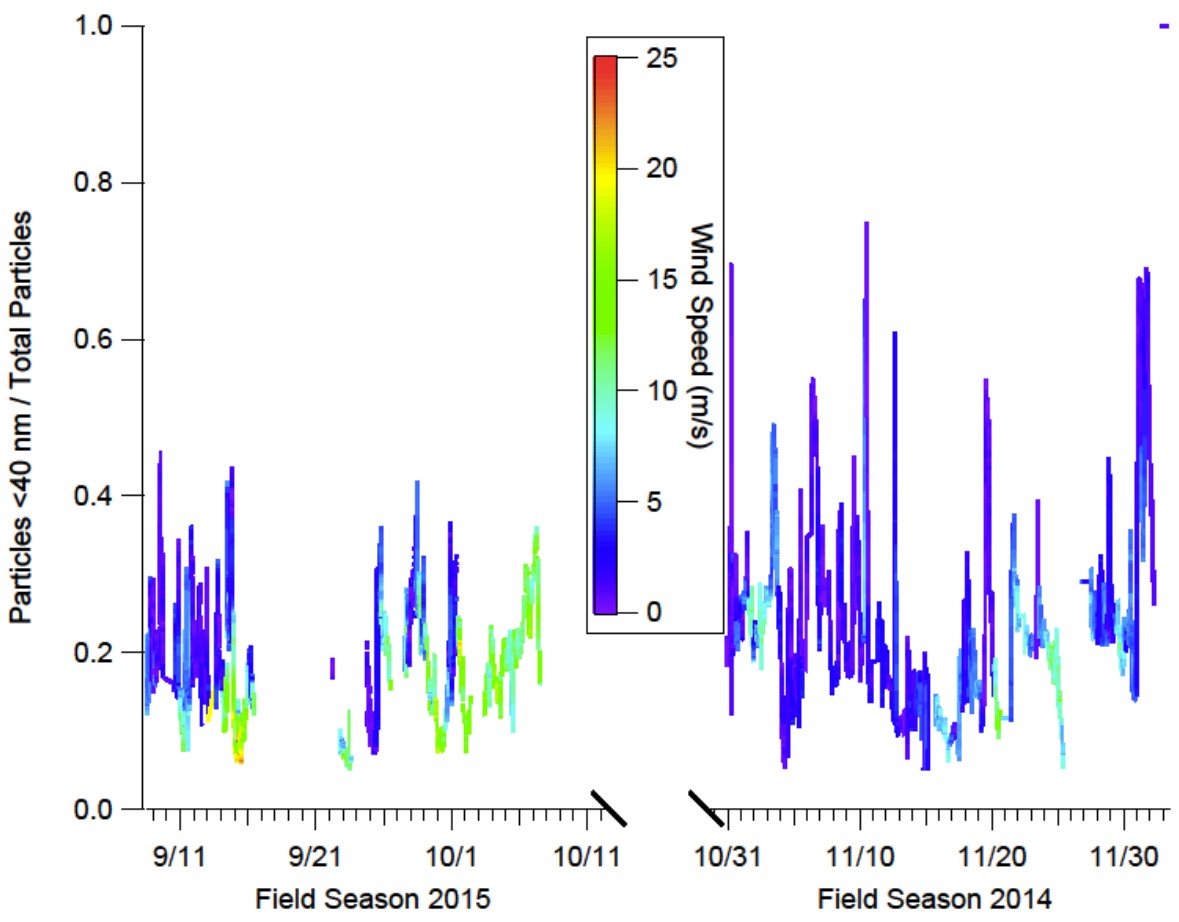

**Figure A2: Fraction of particles less than 40 nm of the total counts from the particle sizing instruments for the the 2015 (SMPS) and 2014 (SEMS) field seasons. Both records are colored by the wind speed (m/s) records.**

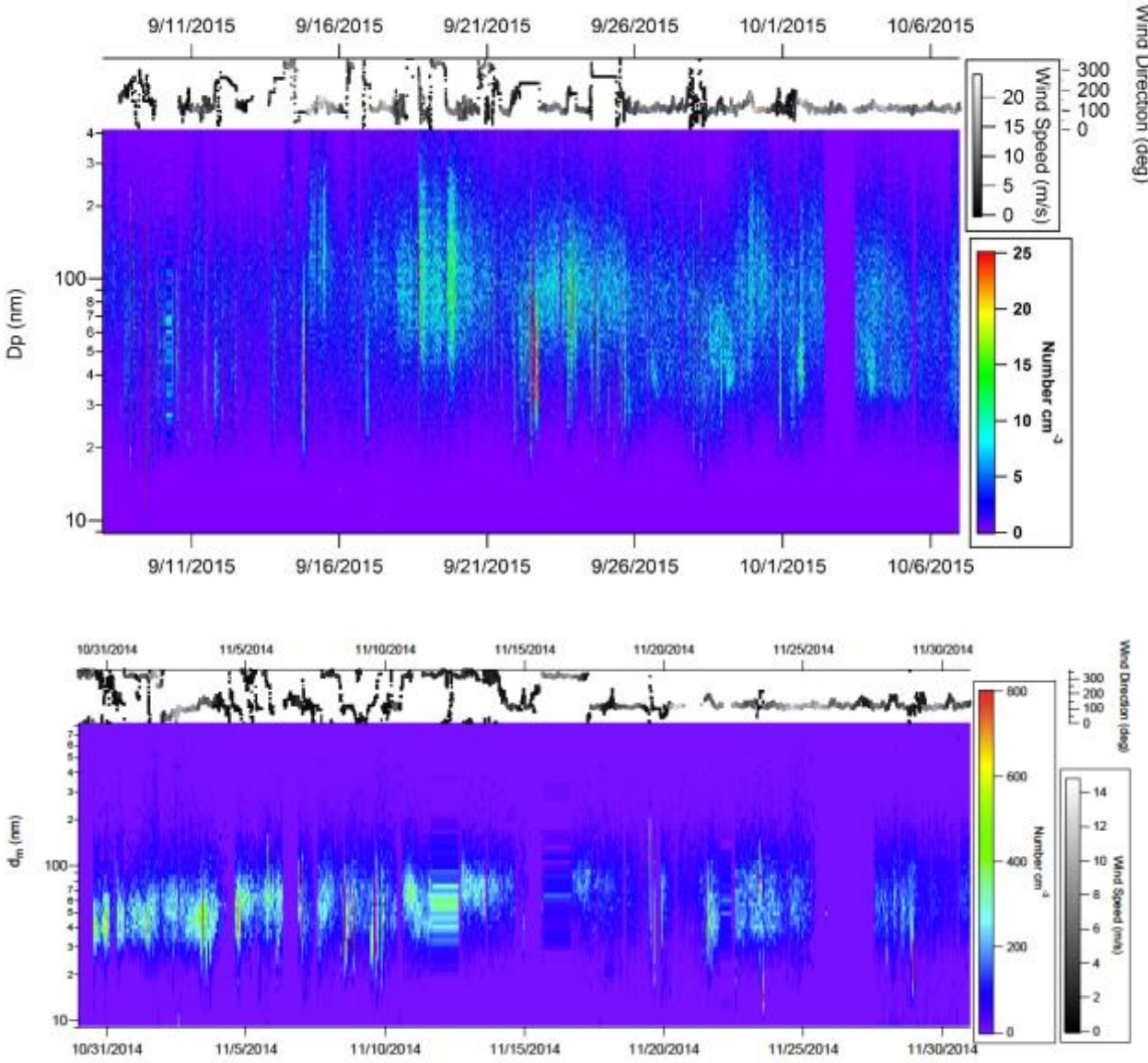

**Figure A3: dN/dlogDp image plots from the particle sizing instruments for the 2015 (a, SMPS) and 2014 (b, SEMS) field seasons. Included are the wind speed and direction records. Note that small (20-30nm) particle enhancements only happen during calm wind** 5 **periods that coincide with elevated *m/z* 55 concentrations in the AMS (not shown) indicating generator contamination.**

### B.1 The transitional aerosol as a refractory number population

Completely ruling out the excess particle counts being due to refractory particles (explanation 1) is impossible with the instrumentation available during 2ODIAC. However, sea-spray, the volcano Mt. Erebus, and the dry valleys are the only major

sources of refractory particles in the region. Since there were no eruptions or seasonally dependent change in volcanic activity during 2ODIAC, Mt. Erebus can likely be ruled out as the source of the particle counts. Sea spray is more difficult to rule out, especially considering the relatively closer ice edge in the spring field season. However, sea spray is predominantly super micron and significant enhancements in the super micron number distribution would lead to significant enhancements in the mass and volume loadings. Such enhancements, however, were not observed in other instrumentation (e.g. Lighthouse OPC).

Further, sea spray source strength is largely determined by wind speed (Madry et al., 2011), and the slow increase observed is unlikely to come from highly variable wind encountered during this time.

### B.2 The transitional aerosol as a non-sulfate number population

The possibility that the AMS is measuring the particles associated with the increased counts of the EPC but that the particles are not sulfate (explanation 2) should also be explored. If non-sulfate particles are the same size or larger than the sulfate

particles, then the ratio of sulfate mass to total non-refractory mass measured in the AMS would likely exhibit opposing trends to Fig. 4. The non-sulfate particles would have to be the same size or larger than the sulfate particles or there could be no observed changed in measured total non-refractory mass. In fact, the ratio of sulfate to total non-refractory mass (both as measured by High Resolution and Unit Mass Resolution in the AMS) is steady for both field seasons when wind speeds are accounted for (Figure A1). Therefore if the particles are of a similar size to the sulfate mode, the observed mass composition

is not enough to explain the trends in Figs. 1 and 4.

### B.3 The transitional aerosol as a population outside the bounds of the AMS or not producing a measureable size distribution signal

The possibility of particles larger than the AMS size cutoff (1 μm aerodynamic diameter, explanation 3) explaining Phase (2) is unlikely due to the inlet geometry. However, the existence of small particles, either significantly smaller than the ~250 nm

mode of sulfate particles or particles smaller than the AMS cutoff (40nm aerodynamic diameter), explaining the trends is possible. If the particles are sulfate, and measureable by the AMS bulk composition, it is not necessarily true that they will produce a measureable signal in the size distribution. Since the AMS is sensitive to mass and not number, small diameter particles do not produce as much signal as a large particle (as mass signal is proportional to $d^3$). Additionally, the use of a 2% chopper to make the sizing measurements, cuts total signal in the sizing mode by a factor of 25 times compared to MS signal

which switches between total particle signal and background signal each for half of the measurement time. Finally, the signal in the sizing mode is spread out over multiple size bins making detection above the instrument baseline noise much more

difficult in sizing mode, and especially challenging in a pristine environment such as Antarctica. Consequently small particles in very low concentration are unlikely to produce a size resolved signal above the noise of the instrument.

Though measuring 40-250nm particle enhancements in the AMS is not possible, we can examine the number fraction of particles less than 40nm to see if these particles contribute to the additional particle number measured by the EPC. Using the particle sizing instruments (SEMS and SMPS) counts between ~7-40nm, the importance of particles less than 40nm can be examined. For a spherical particle of unit density (1.0 g/cm$^3$) or a pure sulfate particle with a density of 1.8 g/cm$^3$ the 40 nm mobility diameter cutoff size would be equivalent to a vacuum aerodynamic diameter of 40 nm (unit density) or 72 nm (density 1.8 g/cm$^3$). If the ratio of the total number of particles below 40nm to the total number of particles increases, then more small particles would be counted by the EPC but not by the AMS. However, during both field seasons, the ratio of <40nm to total particles is steady. In 2014, the ratio of <40nm to total counts from the SEMS averages at $0.23 \pm 0.003$ (confidence interval = 0.05, std. dev. = 0.15). In 2015, the ratio of <40nm to total counts from the SMPS averages at $0.16 \pm 0.002$ (confidence interval = 0.05, std. dev. = 0.09). In fact, neither field season exhibits a strong dependence on wind speed (Figure A2). Removing the high wind events has negligible impacts on both the averages and standard deviations. This again suggests that the background sulfate mass population is relatively temporally and geographically invariant.

However, the possibility of correlation between the ratio of the total counts to the AMS sulfate counts (Fig.4) and the <40nm ratio (Figure A2) should also be examined as it suggests particles are being measured by the AMS but are possibly not sulfate. A Pearson correlation value of -0.4 exists between the two ratios for the late Austral winter/early Austral spring field season of 2015. For the late spring/early summer field season of 2014 this value is 0.04. The 2014 lack of correlation implies that changes in the two ratios are unrelated. The 2015 dataset exhibits a slight anti-correlation suggesting that as the AMS/EPC ratio goes down, the number of sub-40nm particles increases. Even when wind speeds above 8m/s are excluded from the correlation (since high wind speeds and particle counts are strongly correlated) the correlation between AMS/EPC and <40nm/total still stands at -0.37. Additionally, the above correlations can be calculated for larger electrical mobility diameters (50nm and 60nm) to account for the unknown electrical mobility/vacuum aerodynamic cutoff of these small particles. The correlation values change less than 15% with these higher electrical mobility cutoffs. These correlation values have two implications: first, that the change in the fraction of sulfate particles in the total aerosol number population in the summer may not be due to small particles and, second, that changes in the percent sulfate in the early spring may be due in part to small (<40nm mobility diameter) particles.

Figure 4 therefore demonstrates that small particles, either relative to the sulfate mode or below the AMS cutoff diameter, form in the early Austral spring (Phase (2)). Unfortunately, given the instrumentation deployed, these particles remain of an unknown composition. No obvious new particle formation events were captured during 2ODIAC (Figure A3) so the increase in small particles is likely due to differing source regions as the air masses that flowed over the field camp changed.