# Peer review of "A missing source of aerosols in Antarctica – beyond long-range transport, phytoplankton, and photochemistry"

_Atmospheric Chemistry and Physics, 2016_

## Referee Comment (RC2) · Anonymous Referee #2 · 11 Aug 2016

Giordano and co-authors provide a very detailed study of sulfur-containing aerosol from a sea-ice site in East Antarctica covering Austral winter to early summer. In two field seasons, a number of state of the art instruments were deployed to study aerosol number concentrations, size distribution and chemical composition primarily with online methods. The results presented are discussed in great detail and novel conclusions on the provenance of Antarctic aerosol are provided.

This study presents very valuable data from an understudied region and can contribute significantly to what is known about Antarctic aerosol. Nevertheless, before publication several points need to be addressed. Among those are a lack of methodological information, a number of vague conclusions and implicit argumentations, as well as the

need to present more details of the positive matrix factorization results.

General comments:

It is not clear why the authors do not show other species measured by the AMS than sulfate. The discussion on the lower particulate sulfate contribution to the total particle number throughout section 3.2 and more specifically related to Fig. 4 would be much more informative if the authors provided the mass fraction of sulfate measured by the AMS in relation to ammonium, nitrate and organics. I suggest including a figure showing either the time series of these species' masses or their fractional contribution. It seems that the authors plan another publication with a detailed discussion on these data, however this manuscript will strongly benefit from showing the general AMS-related results.

With regards to the PMF analysis, using only sulfur containing fragments as input is novel. For that reason, the methodology and the results need to be shown in more detail. Key diagnostics as outlined in Zhang et al. (2011) should be provided in the SI to show the robustness of the solution. The 3 factor solution is not very convincing as it is currently presented. Factors 1 and 2 are very similar and it is not clear why the authors concluded that these factors are not artificially split, especially since an explanation for the aged biogenic source is missing. Stronger evidence must be provided to justify the 3 factor solution. Also, how stable is the instrument's fragmentation at m/z 48 and 64? Is it stable enough not to introduce variability that's picked up by the PMF analysis?

The contribution of new particle formation (NPF) to the observed aerosol needs a clearer discussion. On p. 11, l. 16 it says that no new particle formation events where observed. Does this mean that you conclude only from the literature that NPF is a potential source? If so, this needs be made explicit and less weight should be given to this conclusion as in that case no direct evidence is be available. If you have evidence for NPF, this is not clearly present in the manuscript currently.

Generally, to provide an impression of the geographical sources of the sampled aerosol

and with that the potential source region for NPF, a back trajectory analysis throughout the measurement periods would be very helpful. This could also address the question of how and why there is a transition period and how source regions might change between seasons.

How representative are the two field seasons? It would be very helpful if at least more long-term meteorological data from the nearby station could be presented to show whether wind direction, wind speed, temperatures, solar irradiance etc. are comparable for the intensive observational periods. This is needed to back up the conclusions of the paper regarding the general background aerosol characteristics, the evolution of aerosol characteristics between winter and summer and the potential contribution of NPF.

Specific comments:

The term 'aerosol population' is used very often. Mostly it is unclear whether the authors refer to mass, number, size distribution or chemical composition. Consider replacing the term by more precise terminology.

p. 2, l. 15 consider including information from Hamilton et al. (2014) that the Southern Ocean is one of the few places left on Earth to sample pristine aerosol.

Hamilton, D. S., Lee, L. A., Pringle, K. J., Reddington, C. L., Spracklen, D. V., and Carslaw, K. S.: Occurrence of pristine aerosol environments on a polluted planet, Proceedings of the National Academy of Sciences, 111, 18466-18471, 2014.

l. 19: not clear whether you refer to the size or spatial distribution

l. 33: it is not necessary to discuss volcanic eruptions to such detail. It is sufficient if you state that volcanic eruptions are not important for Antarctic aerosol except for a few instances. Also, while volcanic eruptions can inject large amounts of sulfur species into the atmosphere and are therefore a temporarily limited major source of aerosol, also anthropogenic emissions need to be considered as they are a constant important

source.

p. 3, l. 7: replace generally by "mostly". Sea spray aerosol does have a submicron fraction.

l. 30 an appropriate reference would also be Petters and Kreidenweis, 2007.

p. 4, l. 23: How did you identify the data? What thresholds did you apply, for example? How many % of the data were that?

p. 5, l. 2f: What is the lower cutoff of the EPC?

L3: if the SP AMS is also high resolution that should be mentioned here explicitly.

l. 6: it is not clear how heating the inlet prevent the collection of wind blown snow. A more precise description is needed.

In the methods sections information is missing on how you verified that the CPCs counted correctly and are comparable for both field seasons. There are also no details on the AMS calibrations and how well did the quantification work? This is important since you compare the sulfate size distribution scaled to the mass measured by the AMS with the size distributions from other instruments to quantify the fraction of sulfate particle to total particle number. How accurate is this. There is no discussion on the potential uncertainties of this comparison (Fig. 4).

l. 32: The text says 15-minute while the figure caption says 1-hour. Please check.

p. 6, l. 6ff: Is the minimum of 50 p/ccm an absolute minimum or did you define concentrations below a percentile as minimum / background? I suggest to work with explicitly percentiles to make your statement more robust. Why is the background concentration higher in spring (125 p/ccm) than in summer (50 p/ccm)?

l. 13f: "Aerosol sulfate mirrors the total aerosol counts..." this statement is too strong. If it "mirrored" the total counts the time series would co-vary more closely or the actual correlation would be higher.

l. 14 ff: provide the months you refer to.

l. 24f: I do not understand the logic of the argument. Please make it explicit what you mean. What is the threshold you refer to? What is it based on? How does it relate to other observations?

l. 28: Why would any enhancement have to be a separate aerosol population? For example, if a major source of aerosol is wave breaking why would stronger winds not lead to an enhancement of the same aerosol population. This argument needs to be revised or clarified.

l. 30f: Which sulfate species are you referring to? Inorganic sulfate or do you include MSA as well? Do you refer to background or enhanced concentrations?

Replace the expression "PToF size" by "vacuum aerodynamic diameter".

p. 7, l. 1: remove "remarkably" here and elsewhere. This is a value judgment.

l. 1f: provide the modal diameters from the Zorn et al. and Schmale et al. references. Which diameter was used in the Jourdain and Legrand publication? Is it comparable with your AMS data?

l. 18: it is not true that in "the rest of the world" external aerosol does not play a role. Further down on this page you describe yourself that external aerosol mixtures occur in urban aerosol. Revise this section here.

p. 8, l. 1f: Explain what you mean by special case? Do you mean that normally there are enough compounds in the atmosphere that condense on pre-existing particles to form internally mixed aerosol at locations where long-range transport is the major source of aerosol? In addition, you make a very general statement about Antarctic aerosol, but you measured on sea-ice actually not very far from the coast. I suggest making your conclusion more relative.

l. 3 – 5: It is unclear to me what you try to say with this and I do not understand what

you base your arguments on.

l. 3.2: Why do you say "If the externally mixed sulfate ... is, ..., the primary component...". As mentioned in the general comments, you have information on other species like ammonium, nitrate and organics from the AMS measurements. You can also estimate the fraction of seasalt that you see with the AMS, see Ovadnevaite et al. 2012.

Ovadnevaite, J., Ceburnis, D., Canagaratna, M., Berresheim, H., Bialek, J., Martucci, G., Worsnop, D. R., and O'Dowd, C.: On the effect of wind speed on submicron sea salt mass concentratio and source fluxes, J. Geophys. Res., 117, 2012.

p.9, l. 2: not clear if you mean sea-salt sulfate or non-sea salt suflate with "not nss-"

p. 10, l. 3f: I do not understand the logic of the argument. Make it explicit. Why would the observed behavior be opposite?

l. 11: What do you mean by inlet dynamics?

p. 11, l. 11: remove important, and make the statement more relative: the correlation is weak, this needs to be reflected in the conclusion.

l. 16: Do you mean that NPF were not observed at all, or that you did not observe any local events but rather already grown particles from NPF further away? As indicated above, the observations and conclusions regarding NPF are not clear.

l. 32: replace "is generating" by "may generate"

l. 33: rephrase to "and it is possible that contributions from" and it is not clear what you mean by non-sulfate aerosol formation mechanisms?

p. 13: more accurate would be : " in regions where the origin of particulate sulfate was dominate by open ..." since there were other major local sources of aerosol as well.

l. 9-11: This sounds like a contradiction to me: the observed concentrations of MSA in

the literature are higher than observations in this study while the argument is the other way around.

l. 30 f: More discussion on the origin of the aged biogenic factor is needed. How can it occur at the earliest time in the season that was observed? What can be the source during winter, when it is dark and more sea-ice is covering the ocean? Also provide number on how many % each factor contributes.

l. 30 Do you mean spring rather than fall?

Conclusions: p. 15, l. 10: Also exploring NFP over the Southern Ocean is important.

Figures:

I suggest including a map or preferably a satellite image showing your measurement sites and the sea-ice extent during the field seasons.

Fig. 1b: There are points below the minimum line. How can that be?

Fig. 2: One needs to guess which line is ESE/NW, low wind and high wind. Consider using a different line type. Why are ESE, Med. Wind and NW, high wind so smooth compared to the other lines?

Fig. 4: How did you smooth? Rename the y-axis to: "sulfate particle number ratio"

Fig. 5: The colors of the symbols from Zorn et al. and Schmale et al. are misleading. As far as I understand the color is not related to the color code. However the colors are part of the range of colors in the code. Either chose different colors for the literature data or use simply open symbols with black margins.

Fig. A3: are the read lines in the lower panel real data?

Technical comments:

p.1, l. 14: low "temporal" resolution and remove the expression in parenthesis which is not needed for the abstract

l. 15: to answer the question about "the chemical composition of" Antarctic aerosols.

l. 16: replace populations by those

l. 18 remove populations

l. 20 the abbreviation SP-AMS does not follow from high resolution. . .

l. 22 "and its evolution in Austral Spring"

be consistent with capitalizing the seasons

l. 23: remove to rest of the aerosol population

l. 26: what are highly aged sulfate particles?

l. 27 & 28 replace population by mass

p. 2 l. 8: "climate impacts depend on their effects on the radiative balance which are a function of the aerosol hygroscopicity, chemical composition and physical optical properties. . ."

l. 10, remove pathways

l. 21f: remove the sentence starting with "Aerosol measurement. . ."

l. 27: remove "of the aerosol population"

l. 28: "the sulfate aerosol mass which has long. . ."

l. 30: shouldn't it be "Kulmala" et al. 2002?

p. 3, l. 9: remove population

l. 25: "of aerosol physical properties. . ."

l. 28: introduce the abbreviation CCN and use it in the next line.

p. 4, l. 3: replace exaggerated by overestimated

l. 7f: "this manuscript focuses on the…."

l. 16: remove the sentence "Cracks in the …" the context is already well enough explained.

l. 22: what does down sampled mean? Is it averaged?

l. 18: remove "but the same order of magnitude as"

p. 8, l. 14: what do you mean by "middle ground"?

p.9, l. 11: replace " is drowned" by "decreases"

l. 13: a subject and verb are missing in this sentence

p. 10., l. 2f: Delete the first sentence, it does not provide any new information.

p. 12, l. 7: data are

l. 18f: delete the first sentence of the sub section, it does not provide any new information.

l. 21: what do you mean by "mirabalite fractionation"?

Fig. 1A: replace "fraction" by "percent"

––––––––––––––––––––––––––––––––

---

## Author Comment (AC1) · 22 Sep 2016

The Authors would like to thank the reviewer for their constructive comments. Specific replies to each comment, and associated changes to the manuscript, are presented here.

**Reviewer 1**: General comments: In several sections, "strong relationships" or "strong function of" are mentioned between two parameters, but it is not necessarily clear in the figures. For instance, on the top of page 6, the authors discuss a relationship between aerosol number and wind. What are the correlations between these two? When looking at both years in Figure 1, it is not immediately clear that this relationship exists until one examines the figure closely, when looking at the 2-minute data. Providing some sort of correlation coefficient would literally strengthen these statements.

Authors: The Pearson correlation coefficient for total aerosol number (from the EPC) to wind speed have been added to the manuscript as per the reviewer's suggestion.

New text: *"Fig. 1b shows two facets of Antarctic aerosols: first, aerosol number concentrations are a function of wind speed (Pearson correlation value of 0.32) and, second, there is a steady-state aerosol concentration during calm and low-wind periods."*

**Reviewer 1**: Presenting more than simply the sulfur species would make the case stronger that sulfate is the major contributor to the AMS aerosol population. What was the percentage of sulfate relative to total AMS particle mass? Specifically, on page 6, line 10, what percentage of the particles measured by the AMS were combustion-derived OA? On page 7, lines 3-4, showing the size distribution of the other AMS types would support the authors' statement here. Without showing the other species, this leaves one to wonder if other aerosol types were relatively high at any point in time (i.e., no graphical evidence provided) in addition to how much of the aerosol were actually sulfate. For context, it would be helpful to provide data on the total aerosol population, perhaps as a time series and size distribution of the relative aerosol types for each season, even if it would be placed in the supporting information.

Authors: We agree with the reviewer that a full discussion of the overall particle composition is important, and we have a manuscript in preparation discussing this in detail. We feel the focus of this paper on sulfate is warranted for the following reasons: 1.) historically the sulfate aerosol population has been of specific scientific interest with regards to the Antarctic aerosol population (e.g. understanding the variability of non-sea-salt sulfate), and 2.) in terms of the aerosol number (not mass) population sulfate aerosol is a key contributor. The open questions regarding the sources, transport, and processing of sulfate over Antarctica are important enough to warrant a paper dedicated to those questions.

We agree with the reviewer that some information contextualizing the sulfate aerosols in terms of the total aerosol is important. Sulfate is the third most abundant species after Cl and Na which is consistent with the literature (approximately 60-80% Na and Cl, 5-30% sulfate depending on wind regimes). Combustion-derived OA was generally not observed except in certain low-wind circumstances and those local emission events have been filtered from this analysis. These details have been added to the text as per the reviewer's suggestion.

New Text: *"While aerosol sulfate is the main focus of this manuscript, it is not the only aerosol component and the relative amount of sulfate measured by the AMS should be contextualized. Over both field seasons, sulfate generally makes up more than 50% of the total mass of the traditionally reported*

*non-refractory species (organics, sulfate, nitrate, and ammonium). Both the absolute amount and relative percentage of total mass of sulfate is higher in 2014 than 2015. Ammonium, organics, and nitrate, in that order, make up the rest of the non-refractory species measured by the AMS. When adding measurements of refractory Na and Cl to the non-refractory species, sulfate is the third most abundant species at 5-30% of the total sub-micron aerosol mass."*

**Reviewer 1**: Several conclusions of the general seasonality of Antarctic aerosol are built upon the observations here, which only span a month or two during two consecutive years. How do the authors know if what they observed was typical or anomalous? For instance, the bottom of page 8 presents broader conclusions based on the intensive measurements presented. These statements would be more convincing if the same month or transition season was measured at least twice, for instance, if both time periods were measured in 2014 and 2015, which obviously cannot be done at this point. Although the observations are very intriguing, the authors should take care in how they interpret the results and try to steer away from making such bold conclusions of what the typical behavior of the aerosol would be this time a year. This could be alleviated by either referring to the observations from the 2014 or 2015 sampling of the transition seasons (versus the transition season in general) or providing more background on previous measurements that would corroborate their observations.

Authors:  A note about the limited duration of the measurements has been added and the transition season has been noted as our "observed" transitional season. Unfortunately, this is the first observation of the transitional phase and previous corroborating measurements do not exist in the literature. The lack of previous observations is due to this being the first deployment of a high-resolution, high-sensitivity aerosol instrument to the continent. These measurements provide evidence of when and where future campaigns should look for non-sulfate particle formation sources and mechanisms.

Added text: *"The results presented here, although limited in seasonal coverage and duration of sampling, suggest that radiative forcing models for Antarctica should continue to treat the sulfate population as an external mixture."*

**Reviewer 1**: It is great that the authors provided such a detailed explanation on the possible sources of uncertainty or limitations in the measurements that could lead to what was observed (i.e., section 3.2), however, this lengthy discussion draws away from the focus on the uniqueness of the observations. Instead, the authors could condense this section to a paragraph or two (and put some or all of the "A" figures in the supporting information), and focus more on bolstering what was observed, particularly the chemistry measurements. Present each of the three explanations separately and more directly, but focus more on the observation itself than what could be wrong with it. As is, when the three possible explanations in the beginning of the section are posed, I thought to myself, they have the data to prove this. Then, the data would be discussed much later. The section is presented more as a thought process to understand the results than a results and discussion section. Also, this section initially is focused on phase 2, but during the explanations, all time periods are discussed. Overall, the section could use some restructuring and condensing, which would provide clarity as well.

Authors: The authors thank the reviewer for this comment and have taken the reviewer's advice. The bulk of the discussion has been moved to an Appendix (Appendix B) and Section 3.2 has been condensed to contain only the conclusions of the section. We believe this enhances the readability of the

manuscript as a whole without losing any detail for those readers who wish to delve into the minutia of the reasoning behind the conclusions presented.

**Reviewer 1**: Perhaps the biggest issue in this manuscript revolves around the new particle formation discussion: The authors provide contradicting evidence that new particle formation is a large contributor to the aerosol number. This is concluded in the abstract, and several locations throughout the manuscript (e.g., page 11, lines 31-32), yet on page 11, lines 16-17, the authors directly state no new particle formation events were captured during 2ODIAC. Please be clear throughout on if new particle formation was a major source. It is difficult to discern any "banana plots" in Figure A3, so where did the conclusion that new particle formation is the major source of aerosol during this time period originate from? Perhaps zooming in on some of those growth events towards the end of 2014 would elucidate if these were indeed new particle formation events or simply emission of small, primary particles.

Authors: The reviewer makes the salient point that "new particle formation" has been used in two different ways: first, to mean local observable particle growth, and second, to refer to the population of unknown composition (newly formed particles) that appears during the transitional period (phase 2). This has been clarified in the text by changing the terminology to refer to "newly formed particles".

New Text: *"…Phase (2) is consistent with measuring newly formed particles that have been transported to our measurement location during a transitional period during the extended Antarctic sunrise…"*

As per the reviewer's comments on Fig. A3: the growth events that are observed (e.g. 3 or 4 times in 2014) are strongly indicated to be contamination from the diesel generators running the field site (as mentioned in the caption of Fig.A3). These periods have been eliminated from the rest of the data presented in the paper and will be removed from this figure as well to prevent any reader confusion.

**Reviewer 1**: More explanation and background is warranted in the PMF section. Are these typical AMS particle classifications that have been previously used or are universal? What are some previous studies that have classified AMS particle types like these? More supporting evidence is needed regarding the classifications for what the particle types were. Labeling the peak fragments in Figure 6 would help as well.

The authors briefly mention the collection of filters for offline analyses in the methods. If the analyses, whatever they might have been, were conducted, those results could provide significant supporting evidence to the conclusions drawn based on the AMS and number concentration measurements that are discussed. Of course, this is also limited by the filter pore size, which was not mentioned. If chemical analytical techniques were applied to the filters, that information could fill in quite a few gaps throughout the manuscript and would potentially provide explanation for much of section 3.2.

Authors: Additional discussion expanding on the PMF section has been added as per the reviewer's suggestion. The use of only sulfur compounds in PMF is novel and comparison to other studies is not possible. Still, some text contextualizing these results has been added.

New Text: "…*Since using only the sulfur containing ions in PMF analysis is novel, it is difficult to compare these PMF results to previously published results. The closest related study is Schmale et al. (2013) which measured Antarctic/Southern Ocean air masses. In both the results presented here and in Schmale et al. (2013), the percent contribution of MSA to the total aerosol burden increases as sunlight (phytoplankton activity) increases over the Southern Ocean. Additionally, the MSA associated factor in that study is*

*postulated to contribute significantly to the total sulfate signal, although it is not measured explicitly, which agrees with the results here…."*

Secondly, the filters have recently been analyzed using an IC and the results will be discussed in an upcoming manuscript. Broadly speaking, the filter results are consistent with the AMS data, although direct comparison with the AMS data is not possible due to differences in size cuts on the filters (which include supermicron particles) to the submicron measurements of the AMS. The supermicron mass signal from the filters is dominant and they integrate over far longer time scales. For these reasons the filter data would not add significantly to this manuscript, and could distract from the central focus.

**Specific comments**:

**Reviewer 1**: Page 5, line 4: Although there are a couple references provided, a few sentences on the specifications and operating principles for the SP-AMS is needed. Especially considering the authors discuss the instrument limitations later on in the manuscript.

Both reviewers have noted that it is unclear that the SP-AMS is largely the same instrument as HR-ToF-AMS with the addition of an extra laser to measure black carbon. This extra laser was not utilized during this campaign, making the SP-AMS identical to the HR-ToF-AMS in terms of operating principle and performance. This has been noted in the text to prevent confusion by readers.

New Text: *"…aerosol composition was measured with a Soot Particle Aerosol Mass Spectrometer (Aerodyne Research Inc. Billerica, MA, SP-AMS, DeCarlo et al., 2006; Onasch et al. 2012). The SP-AMS is a combination of the Aerodyne High-Resolution Time-of-Flight aerosol mass spectrometer (HR-ToF-AMS) and a soot vaporizing laser (from Droplet Meas. Tech.)."*

**Reviewer 1**: On page 6, line 31, a PToF size is discussed but it us unknown up to this point that the SP-AMS contains a ToF mass spectrometer. Defining this in the methods would alleviate any confusion.

Authors: This has been clarified in the text via the addition pointed out in the previous comment.

New Text: *"…The SP-AMS is a combination of the Aerodyne High-Resolution Time-of-Flight aerosol mass spectrometer (HR-ToF-AMS) and a soot vaporizing laser (from Droplet Meas. Tech.)."*

**Reviewer 1**: Page 5, line 5: What offline analyses?

As the filter data are not directly relevant and are beyond the scope of this manuscript, this line has been removed from the text.

**Reviewer 1**: Page 5, lines 29-30: Is this typical and/or expected in this region?

Authors: Both the bimodal wind direction distribution and higher late-winter/early-spring wind speeds are typical of the region. A reference to an analysis of the prevailing meteorology of the Ross Island region has been added to the manuscript.

**Reviewer 1**: Page 5, line 32: The caption for the figure says 1-hour, not 15-minute.

Authors:  The text line has been changed to reflect the (correct) figure caption.

New Text: "*Figure 1b shows the number concentration from the EPC over both field seasons. The figure shows the 2-minute average as well as a 1 hour average.*"

**Reviewer 1**: Page 7, line 16: Use the acronym for CCN when they are first discussed in the introduction and simply use the acronym here (it was spelled out twice in the introduction).

Authors:  The text has been changed as per the reviewer's suggestion.

New Text: "*…radiative forcing of Antarctic aerosol and in predicting CCN number concentrations ($N_{CCN}$) in the Antarctic troposphere.*"

**Reviewer 1**: Page 7, lines 20-22: This statement is highly speculative based on the data provided. Considering the limitations of the AMS (refractive aerosol, the size range), this conclusion is not fully supported by the available observations, especially since the measurements were not conducted during all seasons (Sep – Nov). A statement of this level would require a longer time period of measurements covering a wider range of aerosol types and sizes.

Authors:  The text has been modified regarding the seasonality and limited nature of the 2ODIAC measurements.

New Text: "*The results presented here, although limited in seasonal coverage and duration of sampling, suggest that radiative forcing models for Antarctica should continue to treat the sulfate population as an external mixture. This work does support the assumptions of older estimates of radiative forcing for sulfate aerosols over Antarctica of approx. -0.1 $Wm^{-2}$ (Myhre et al., 1998).* "

**Reviewer 1**: Page 10, line 30: Wind speed is all that is presented here, not all local meteorology. Simply stating wind speed would suffice.

Authors: The text has been changed as per the reviewer's suggestion.

New Text: "*In fact, neither field season exhibits a strong dependence on wind speed…*".

**Reviewer 1**: Figures: I get why the authors are showing 2015 before 2014 in the figures, to enable the data to be presented in a seasonal versus chronological order. Perhaps labeling them as "Austral spring (2014)" and Austral summer (2015)" would make more sense if keeping the data in this order.

Authors: The figures have been modified as the reviewer suggests.

**Reviewer 1**: Figure 1: What are the time resolutions for wind direction/speed and AMS?

Authors: The figure caption has been modified to reflect the time resolutions.

New Text: "*Figure 1: For both the 2014 and 2015 field seasons, with 2015 leftmost: A) Wind direction record colored as a function of wind speed, displayed as a 2-minute average record B) 2-minute (light blue) and 1-hour (black) records of particle number concentration from the EPC, C) 2-minute records of sulfate concentration from the aerosol mass spectrometer. Dotted lines on (B) indicate the minimums in particle number concentrations (99$^{th}$ percentile) measured over the field seasons.*"

**Reviewer 1**: Figure 3: Why is the UHSAS so noisy? Why is only 2014 shown?

Authors:  The UHSAS is a high speed and high resolution instrument, and for this campaign data was taken at 10 second resolution and then coherently averaged to longer times. During low and medium wind speeds, this instrument is operating close to the detection limit for the short sample times, which can introduce counting noise to the signal. Furthermore, there are some small artifacts introduced to the coherently averaged signal from the combined gain stages within the instrument, however the overall size distribution from the UHSAS is consistent with the data from the other particle-sizing instruments. Unfortunately, the UHSAS was damaged during shipment and was inoperable for the 2015 field season.

**Reviewer 1**: Figure 6: I see this is the combined time series from the different years, yet could cause some confusion since these data were not obtained from the same year. Be sure to label the year that corresponds to each data time period on this figure, similar to the previous figures. Also, labeling the peak fragments on the mass spectra would be helpful.

Authors: The figure has been modified as per the reviewer's suggestion.

---

## Author Comment (AC2) · 22 Sep 2016

The Authors would like to thank the reviewer for their constructive comments and highly detailed proof reading of the manuscript. Specific replies to each comment and associated changes to the manuscript are outlined in this document.

General comments:

**Reviewer 2**: It is not clear why the authors do not show other species measured by the AMS than sulfate. The discussion on the lower particulate sulfate contribution to the total particle number throughout section 3.2 and more specifically related to Fig. 4 would be much more informative if the authors provided the mass fraction of sulfate measured by the AMS in relation to ammonium, nitrate and organics. I suggest including a figure showing either the time series of these species' masses or their fractional contribution. It seems that the authors plan another publication with a detailed discussion on these data, however this manuscript will strongly benefit from showing the general AMS-related results.

Authors: We agree with the reviewer that a full discussion of the overall particle composition is important, and we have a manuscript in preparation discussing this in detail.  We feel the focus of this paper on sulfate is warranted for the following reasons: 1.) historically the sulfate aerosol population has been of specific scientific interest to the Antarctic aerosol population (e.g. understanding the variability of non-sea-salt sulfate), and 2.) in terms of the aerosol number (not mass) population sulfate aerosol is a key contributor.   The open questions regarding the sources, transport, and processing of sulfate over Antarctica are important enough to warrant a paper dedicated to those questions.

We agree with the reviewer that some information contextualizing the sulfate aerosols in terms of the total aerosol is important.  Sulfate is the third most abundant species after Cl and Na which is consistent with the literature (approximately 60-80% Na and Cl, 5-30% sulfate depending on wind regimes). Combustion-derived OA was generally not observed except in certain low-wind circumstances and those local emission events have been filtered from this analysis.  These details have been added to the text as per the reviewer's suggestion.

New Text: "*While aerosol sulfate is the main focus of this manuscript, it is not the only aerosol component and the relative amount of sulfate measured by the AMS should be contextualized. Over both field seasons, sulfate generally makes up more than 50% of the total mass of the traditionally reported non-refractory species (organics, sulfate, nitrate, and ammonium). Both the absolute amount and relative percentage of total mass of sulfate is higher in 2014 than 2015. Ammonium, organics, and nitrate, in that order, make up the rest of the non-refractory species measured by the AMS. When adding measurements of refractory Na and Cl to the non-refractory species, sulfate is the third most abundant species at 5-30% of the total sub-micron aerosol mass.*"

**Reviewer 2**: With regards to the PMF analysis, using only sulfur containing fragments as input is novel. For that reason, the methodology and the results need to be shown in more detail. Key diagnostics as outlined in Zhang et al. (2011) should be provided in the SI to show the robustness of the solution. The 3 factor solution is not very convincing as it is currently presented. Factors 1 and 2 are very similar and it is not clear why the authors concluded that these factors are not artificially split, especially since an explanation for the aged biogenic source is missing. Stronger evidence must be provided to justify the 3

factor solution. Also, how stable is the instrument's fragmentation at m/z 48 and 64? Is it stable enough not to introduce variability that's picked up by the PMF analysis?

Authors: The diagnostics as outlined in Zhang et al. (2011) have been included in the SI as per the reviewer's suggestion. It should be noted that the 2014 data has higher residuals than the 2015 data. The authors believe this is primarily a factor of how the combined solution was constructed: 2014 and 2015 data sets were run separately and similar (though not completely identical) factor solutions were obtained with reasonable residuals. Because the individual solutions appeared reasonable, the 2 data sets were combined and PMF run on the combined data set. In the combined set, over all factors and fpeaks, the residuals for the summer (2014) data are much larger in comparison to the 2015 data. We believe this is due to the instrument performance in 2015 vs 2014. Small changes in instrument background and sensitivity will impact the associated error of the instrument which goes into the PMF solution. In a low-signal environment such as Antarctica, this may cause the observed differences in residual, and influence the mass spectra identified by PMF.

Additional discussion and contextualization of the PMF has also been added to the text as per the reviewer's suggestion.

New Text:

*Despite the minimal contribution of the aged biogenic factor, the three factor solution was chosen over the two factor solution for two reasons. The primary reason is the inadequacy of the 2 factor PMF solution with regard to MSA, which based on previous measurements in the Southern Ocean and the presence of a marker ion ($CH_3SO_2^+$ at m/z 79) should make up some of the sulfur contribution. 2-factor PMF results either apportioned m/z 79 to 2 factors with 48:64 ratios that did not resemble any known substance (e.g. things tested in a lab setting included ammonium sulfate, pure MSA, diluted $H_2SO_4$, and southern ocean sea water) or apportioned majority of m/z 79 to a factor that was not temporally consistent with the $CH_3SO_2^+$ fragment in the dataset. The secondary reason for choosing the three factor solution is that three factors was consistently the number where diminishing returns in $Q/Q_{exp}$ began to occur. The attribution of the MSA marker ion to the aged biogenic and biogenic/MSA factor indicates that both factors are likely representative of either MSA directly or of "biologically influenced" aerosols. Comparison to direct atomization of MSA into the AMS (see SI) suggests that the biogenic factor is made up of more than just MSA contributions since PMF did not find a "pure" MSA factor mass spectra for this dataset. The ratios of $CH_3SO_2^+$ to the major sulfate peaks ($SO^+$, $SO_2^+$, $HSO_3^+$, $SO_4^+$) in the two biogenic factors differ from pure MSA measured by the AMS in the laboratory.*

**Reviewer 2**: The contribution of new particle formation (NPF) to the observed aerosol needs a clearer discussion. On p. 11, l. 16 it says that no new particle formation events where observed. Does this mean that you conclude only from the literature that NPF is a potential source? If so, this needs be made explicit and less weight should be given to this conclusion as in that case no direct evidence is be available. If you have evidence for NPF, this is not clearly present in the manuscript currently

Authors: The reviewer has noted that NPF is used to mean two different things in this manuscript: local observable particle growth and regional (unobserved growth) particle formation. In the case of the former, there were no observed local NPF events. In the case of the latter, section 3.2 goes into detail

about why we believe (regional) NPF and transport to our site is a major factor in the Phase (2) aerosol population. NPF has been clarified in the text, in conjunction with Reviewer 1's comments, to refer to "newly formed particles" when the latter case is meant to avoid confusion.

**Reviewer 2**:  Generally, to provide an impression of the geographical sources of the sampled aerosol and with that the potential source region for NPF, a back trajectory analysis throughout the measurement periods would be very helpful. This could also address the question of how and why there is a transition period and how source regions might change between seasons.

Authors: In addition to the clarification regarding NPF above, we generally agree with the reviewer. However, accuracy of back trajectory calculations are highly dependent on the meteorological data that feeds them. In the case of McMurdo Antarctica, the available meteorology comes from the GDAS 0.5° x 0.5° record. The one-half degree resolution of the data is insufficient to resolve local orography effects on the air flows that arise from the complex topography of the Ross Island Region.

Understanding this limitation, we have performed HYSPLIT back trajectory analyses over the whole of the 2ODIAC campaign. Generally speaking, the majority of air masses were subject to long-range transport over the continent though some air masses did originate over the Southern Ocean. However, without resolving orography, it is impossible to tell if an air mass was exposed to open ocean immediately prior to sampling or originated completely inland.

Beyond being confident that the data presented in this manuscript are not contaminated by anthropogenic sources (McMurdo or Scott Base, 2ODIAC generators, etc), it is not currently possible to identify with any certainty the local source regions observed during the field campaign. This is the reason we are careful to discuss all of the known and suspected particle formation mechanisms relevant to Antarctica (see p.12).

**Reviewer 2**:  How representative are the two field seasons? It would be very helpful if at least more long-term meteorological data from the nearby station could be presented to show whether wind direction, wind speed, temperatures, solar irradiance etc. are comparable for the intensive observational periods. This is needed to back up the conclusions of the paper regarding the general background aerosol characteristics, the evolution of aerosol characteristics between winter and summer and the potential contribution of NPF.

Authors: Both the bimodal wind direction distribution and higher late-winter/early-spring wind speeds are typical of the region, and this has been noted in the manuscript.   Including a detailed climatology for the region is beyond the scope of this work, however a reference to an analysis of the prevailing meteorology of the Ross Island region has been added to the manuscript.

New Text: *"…These meteorological patterns and seasonal differences are not unusual for this region (Seefeldt et al., 2003)."* (p6, l.15)

Specific comments:

**Reviewer 2**: The term 'aerosol population' is used very often. Mostly it is unclear whether the authors refer to mass, number, size distribution or chemical composition. Consider replacing the term by more precise terminology.

Authors: "Aerosol population" has been changed to identify aerosol number or aerosol mass specifically throughout the paper. This has been clarified early in the text to reflect this as per the reviewer's suggestion.

**Reviewer 2**: p. 2, l. 15 consider including information from Hamilton et al. (2014) that the Southern Ocean is one of the few places left on Earth to sample pristine aerosol.

Hamilton, D. S., Lee, L. A., Pringle, K. J., Reddington, C. L., Spracklen, D. V., and Carslaw, K. S.: Occurrence of pristine aerosol environments on a polluted planet, Proceedings of the National Academy of Sciences, 111, 18466-18471, 2014.

Authors: This has been added to the text as per the reviewer's suggestion.

New Text: "…*Measurements in Antarctica, provide insight into one of the more pristine environments and can be useful in the understanding of preindustrial background aerosol (e.g. Hamilton et al., 2014). However, the ability to sample pristine aerosols is directly related to an areas inaccessibility…*"

**Reviewer 2**: l. 33: it is not necessary to discuss volcanic eruptions to such detail. It is sufficient if you state that volcanic eruptions are not important for Antarctic aerosol except for a few instances. Also, while volcanic eruptions can inject large amounts of sulfur species into the atmosphere and are therefore a temporarily limited major source of aerosol, also anthropogenic emissions need to be considered as they are a constant important source.

Authors: Given the proximity of the field site to an active volcano that is constantly emitting aerosols and $SO_2$, we believe that the three sentences explaining why Mt. Erebus is not likely impacting the results presented here are warranted. The mass spectral fingerprint of Mt. Erebus is also more difficult to distinguish from background (without isotopic analysis), unlike the anthropogenic sulfur sources near the field site (e.g. McMurdo, Scott Base, diesel powered energy/transportation).

**Reviewer 2**: p. 3, l. 7: replace generally by "mostly". Sea spray aerosol does have a submicron fraction.

Authors: This has been changed in the text as suggested.

New Text: "…*Sea-spray aerosols are mostly supermicron in size and production is a strong function of wind speed…*"

**Reviewer 2**: l. 30 an appropriate reference would also be Petters and Kreidenweis, 2007.

Authors: This has been added to the text as suggested.

New Text: *"…Determining the CCN spectrum of a given aerosol population is possible once the size distribution, size-resolved composition, and mixing state of the aerosol population is known (Petters and Kreidenweis, 2007…"*

**Reviewer 2**: l. 3 – 5: It is unclear to me what you try to say with this and I do not understand what you base your arguments on.

Authors: The major point is that the extent of external mixtures tends to decrease as distance from emission source increases. This does not appear to be the case for the measurements presented here. The text has been modified to better reflect the meaning.

New Text: "*…Because Antarctic aerosols seem to primarily be composed of sulfates and salts, the effect of the mixing state on cloud forming predictions may be minimized over the continent itself but overestimated as continental air masses flow out over the Southern Ocean and gain organic components…*"

**Reviewer 2**: l. 3.2: Why do you say "If the externally mixed sulfate . . . is, . . ., the primary component. . .". As mentioned in the general comments, you have information on other species like ammonium, nitrate and organics from the AMS measurements. You can also estimate the fraction of seasalt that you see with the AMS, see Ovadnevaite et al. 2012. Ovadnevaite, J., Ceburnis, D., Canagaratna, M., Berresheim, H., Bialek, J., Martucci, G., Worsnop, D. R., and O'Dowd, C.: On the effect of wind speed on submicron sea salt mass concentratio and source fluxes, J. Geophys. Res., 117, 2012.

Authors: As noted earlier, the mass fraction of sulfate has been added to the text. Of the "traditional" AMS species, sulfate makes up the majority of the aerosol measured. Additionally, an estimation of the refractory Na and Cl has been performed (similar to Salcedo et al., 2010) and is the subject of an upcoming manuscript and is beyond the scope of this paper. Even upon including the "non-traditional" species, i.e. refractory sea salts, sulfate is still the third most common species behind Cl and Na.

**Reviewer 2**: p.9, l. 2: not clear if you mean sea-salt sulfate or non-sea salt suflate with "not nss-"

Authors: This has been clarified in the text.

New Text: "*…persistent aerosol sulfate component (total, i.e. not nss-) seen multiple times over the continent in the winter…*"

**Reviewer 2**: p. 10, l. 3f: I do not understand the logic of the argument. Make it explicit. Why would the observed behavior be opposite?

Authors: The text has been clarified with the sentence: *"The non-sulfate particles would have to be the same size or larger than the sulfate particles or there could be no observed changed in measured total mass."*

**Reviewer 2**: l. 11: What do you mean by inlet dynamics?

Authors: Dynamics has been changed to geometry.

**Reviewer 2**: p. 11, l. 11: remove important, and make the statement more relative: the correlation is weak, this needs to be reflected in the conclusion.

Authors: The text has been modified as per the reviewer's suggestion.

New Text: "*…These correlation values have two implications: first, that the change….*"

**Reviewer 2**: l. 16: Do you mean that NPF were not observed at all, or that you did not observe any local events but rather already grown particles from NPF further away? As indicated above, the observations and conclusions regarding NPF are not clear.

Authors: As per the previous comment, NPF has been clarified in the next to read as "newly formed particles" where "regional NPF" was meant in the text.

**Reviewer 2**: l. 32: replace "is generating" by "may generate"

Authors: This has been changed in the text.

**Reviewer 2**: l. 33: rephrase to "and it is possible that contributions from" and it is not clear what you mean by non-sulfate aerosol formation mechanisms?

Authors: This sentence has been modified in the text. Non-sulfate mechanisms refers to the possibility that DMS does not play a role in this aerosol. This has been clarified in the text as well.

**Reviewer 2**: p. 13: more accurate would be : " in regions where the origin of particulate sulfate was dominate by open . . ." since there were other major local sources of aerosol as well.

Authors: The text has been modified as per the reviewer's suggestion.

New text: "…*Both of the previous measurements took place in regions where the origin of particulate sulfate was dominated by open ocean source regions and took place in the austral summer and fall…*"

**Reviewer 2**: l. 9-11: This sounds like a contradiction to me: the observed concentrations of MSA in the literature are higher than observations in this study while the argument is the other way around.

Authors: Lower concentrations in 2ODIAC, not the previous campaigns. This has been clarified in the text.

New Text: *"…Lower MSA and sulfate concentrations during 2ODIAC are therefore not surprising given the differences in season and location as compared to the previous studies…*"

**Reviewer 2**: l. 30 f: More discussion on the origin of the aged biogenic factor is needed. How can it occur at the earliest time in the season that was observed? What can be the source during winter, when it is dark and more sea-ice is covering the ocean? Also provide number on how many % each factor contributes.

Authors: The PMF discussion section has been expanded, including discussion on the origin of the aged biogenic factor. During the winter, the most likely source is long-range transport from areas of the Southern Ocean that are not ice-covered and in perpetual darkness. However, it should be noted that our measurements took place at the extreme end of winter/early spring. There was measureable (~50 W/m2) sunlight for 4-5 hours during the first few days of the 2015 campaign. The sun came up quickly after that. The actual distance one has to travel north from Ross Island to reach "normal" daylight in early September is not that far.

The percent contribution for each factor has been included as well.

**Reviewer 2**: l. 30 Do you mean spring rather than fall?

Authors: The reviewer is correct, fall has been changed to spring.

**Reviewer 2**:Conclusions: p. 15, l. 10: Also exploring NFP over the Southern Ocean is important.

Authors: The text has been modified as per the reviewer's suggestion.

New Text: "*This work further underscores the need to closely examine new particle formation over Antarctica, and the Southern Ocean, in the early Austral spring.*"

**Figures**:

**Reviewer 2**:I suggest including a map or preferably a satellite image showing your measurement sites and the sea-ice extent during the field seasons.

Authors: A satellite image (Landsat 8 SLI, retrieval date 10/14/15) has been added to SI. The 2014 and 2015 sea ice edges and field sites have been marked.

**Reviewer 2**:Fig. 1b: There are points below the minimum line. How can that be?

Authors: Some points are filter periods that were not removed correctly, the minimum is defined as the 99$^{th}$ percentile. The filter periods have now been removed.

**Reviewer 2**:Fig. 2: One needs to guess which line is ESE/NW, low wind and high wind. Consider using a different line type. Why are ESE, Med. Wind and NW, high wind so smooth compared to the other lines?

Authors: The line types have been changed to address the reviewer's concern.

ESE_MW is approximately the same "smoothness" as ESE_HW. These 2 wind regimes, along with NW_HW, had higher mass loadings than the low wind speed regimes which results in a "smoother" trace (as signal:noise is improved at higher mass loadings). NW_MW had similar mass loadings but had a much reduced sampling time as compared to the other three (high signal) regimes. This exacerbates the noise in the NW_MW trace as well.

**Reviewer 2**:Fig. 4: How did you smooth? Rename the y-axis to: "sulfate particle number ratio"

Authors: "Boxcar smoothing" has been included in the text. The figure axis has been changed as per the reviewer's request.

**Reviewer 2**:Fig. 5: The colors of the symbols from Zorn et al. and Schmale et al. are misleading. As far as I understand the color is not related to the color code. However the colors are part of the range of colors in the code. Either chose different colors for the literature data or use simply open symbols with black margins.

Authors: The Zorn and Schmale data symbols have been changed to black and grey to resolve the issue the reviewer points out.

**Reviewer 2Reviewer 2**: Fig. A3: are the read lines in the lower panel real data?

Authors: The red lines were an AMS IE calibrations/size calibrations that did not get removed from the figure. They are now removed. As per reviewer 1, the generator contamination has also been removed from the figure.

Technical comments:

**Reviewer 2**: p.1, l. 14: low "temporal" resolution and remove the expression in parenthesis which is not needed for the abstract

This has been changed in the text as suggested.

**Reviewer 2**: l. 15: to answer the question about "the chemical composition of" Antarctic aerosols.

This has been changed in the text as suggested.

**Reviewer 2**: l. 16: replace populations by those

"Populations" keeps the sentence completely unambiguous. "Those" could conceivably refer to "seasonal cycles". This has not been changed to prevent ambiguity.

**Reviewer 2**: l. 18 remove populations

This has been changed in the text as suggested.

**Reviewer 2**: l. 20 the abbreviation SP-AMS does not follow from high resolution. . .

The SP-AMS is an upgrade to the HR-ToF-AMS (). This has been clarified in the instrumentation section.

**Reviewer 2**: l. 22 "and its evolution in Austral Spring" be consistent with capitalizing the seasons

The capitalization of the seasons has been made consistent across the entirety of the manuscript.

**Reviewer 2**: l. 23: remove to rest of the aerosol population

This has been changed in the text as suggested.

**Reviewer 2**: l. 26: what are highly aged sulfate particles?

"Aged sulfate" is defined in the text, specifically the PMF section.

**Reviewer 2**: l. 27 & 28 replace population by mass

The first instance has been changed as suggested, the second instance has been left as population.

**Reviewer 2**: p. 2 l. 8: "climate impacts depend on their effects on the radiative balance which are a function of the aerosol hygroscopicity, chemical composition and physical optical properties. . ."

**Reviewer 2**: l. 10, remove pathways

This has been changed in the text as suggested.

**Reviewer 2**: l. 21f: remove the sentence starting with "Aerosol measurement. . ."

This has been changed in the text as suggested.

**Reviewer 2**: l. 27: remove "of the aerosol population"

The sentence has been reworded to "…component of that aerosol, especially…" This prevents the ambiguity and improves flow of the sentence.

**Reviewer 2**: l. 28: "the sulfate aerosol mass which has long. . ."

This has been changed in the text as suggested.

**Reviewer 2**: l. 30: shouldn't it be "Kulmala" et al. 2002?

The authors thank the reviewer for catching this typo.

**Reviewer 2**: p. 3, l. 9: remove population

This has been changed in the text as suggested.

**Reviewer 2**: l. 25: "of aerosol physical properties. . ."

This has been changed in the text as suggested.

**Reviewer 2**: l. 28: introduce the abbreviation CCN and use it in the next line.

This has been changed in the text as suggested.

**Reviewer 2**: p. 4, l. 3: replace exaggerated by overestimated

This has been changed in the text as suggested.

**Reviewer 2**: l. 7f: "this manuscript focuses on the. . .."

This has been changed in the text as suggested.

**Reviewer 2**: l. 16: remove the sentence "Cracks in the . . ." the context is already well enough explained.

This has been changed in the text as suggested.

**Reviewer 2**: P5 l. 22: what does down sampled mean? Is it averaged?

Down sampled has been changed to averaged in the text.

**Reviewer 2**: l. 18: remove "but the same order of magnitude as"

This has been changed in the text as suggested.

**Reviewer 2**: p. 8, l. 14: what do you mean by "middle ground"?

This has been changed in the text to average value.

**Reviewer 2**: p.9, l. 11: replace " is drowned" by "decreases"

This has been changed in the text as suggested.

**Reviewer 2**: l. 13: a subject and verb are missing in this sentence

"From Fig. 1 during Phase (2), both the total counts on the EPC and the sulfate mass in the AMS trend upward but total counts increases faster than the mass captured in the AMS."

To Trend is the verb in this sentence, EPC counts and sulfate mass is the subject. "But total counts…" has been revised to a second sentence.

**Reviewer 2**: p. 10., l. 2f: Delete the first sentence, it does not provide any new information.

This section has been revised completely as per Reviewer 1's suggestions.

**Reviewer 2**: p. 12, l. 7: data are

This has been changed in the text as suggested.

**Reviewer 2**: l. 18f: delete the first sentence of the sub section, it does not provide any new information.

This section has been revised completely as per Reviewer 1's suggestions.

**Reviewer 2**:  l. 21: what do you mean by "mirabalite fractionation"?

Mirabalite is defined in the introduction as is the fact that sodium fractionates during its formation.

**Reviewer 2**: Fig. 1A: replace "fraction" by "percent"

This has been changed in the figure as suggested assuming the reviewer means Fig. A1.

---

## Referee Report (RR1)

Giordano et al. have done a fine job revising the manuscript based on the reviewer comments. Although they provided adequate responses to the comments, some responses did not result in a change to the manuscript, and there are a few additional issues that need to be addressed prior to publication in ACP.

**General comments:**

The main concern I have is regarding the AMS results, namely determining the size of the sulfate aerosol. I am no expert when it comes to AMS, but doesn't the AMS only measure single particles when there is sufficient mass in each particle? Jayne et al. (2000) report on the AMS, "...the current sensitivity of this AMS for single particle counting with 100% efficiency is $\sim 2$ x $10^{-14}$ gm ($\sim 300$ nm for a pure component particle) and the sensitivity for the signal averaging mode is $\sim 0.25$ µg m$^{-3}$ for several minutes of signal integration." Considering the authors report sulfate aerosol sizes of < 300 nm ($\sim 250$ nm), wouldn't the AMS need to integrate and average > 1 particle? Further, according to Onashe et al. (2012), "The SP-AMS, like other AMS instruments, provides average mass spectrometric measurements of the ensemble of sampled aerosols rather than measurements on a particle-by particle basis." In light of this information, the authors should be clear on how the size of specifically sulfate aerosol was delineated from the measurements. Was sufficient mass present in each particle to actually measure the size *and* composition of single particles? If a bulk sample was measured, how can the authors be sure the sizes measured were purely sulfate particles? Further, the authors focus on differentiating internal versus external mixtures, which is indeed important. Although I am happy to see this distinguished, because the methods regarding how single particles were measured are not described in the level of detailed needed, it is not clear how the authors can differentiate external mixtures. If AMS integration and averaging for > 1 particle was done, how can external versus internal mixing state be determined? Perhaps this concern originates from my naïve knowledge of the AMS, but the authors should at least clarify how they came up with the size and composition of single particles to delineate sulfate aerosols.

T. B. Onasch , A. Trimborn , E. C. Fortner , J. T. Jayne , G. L. Kok , L. R. Williams , P. Davidovits & D. R. Worsnop (2012) Soot Particle Aerosol Mass Spectrometer: Development, Validation, and Initial Application, Aerosol Science and Technology, 46:7, 804-817, DOI: 10.1080/02786826.2012.663948.

J. T. Jayne , D. C. Leard , X. Zhang , P. Davidovits , K. A. Smith , C. E. Kolb & D. R. Worsnop (2000) Development of an Aerosol Mass Spectrometer for Size and Composition Analysis of Submicron Particles, Aerosol Science and Technology, 33:1-2, 49-70, DOI: 10.1080/027868200410840.

It doesn't appear that the results support the main conclusion that sulfate is the dominant Antarctic aerosol, at least, not at this level of breadth. Figure A1 shows the percent of sulfate to total AMS mass, but the percentage is < 40% most of the time, and doesn't seem to equate to the average value the authors present in the text on P 7 l 8-9, "Over both field seasons, sulfate generally makes up more than 50% of the total mass of the traditionally reported non-refractory species (organics, sulfate, nitrate, and ammonium)." Further, the authors affirm that this value is even less when considering refractory Na and Cl (5-30%). This also does not include other non-refractory species like mineral dust. By number, I can see how this conclusion holds true (i.e., based on Fig. 4). A few things here: (1) Where did the Na and Cl come from? How were these measured? (2) Based on the concern regarding how sulfate number was derived, how can this hold true? (3) If sulfate particles were indeed measured on a particle-by-particle basis, thus size could be measured by AMS, the authors should

provide a more explicit conclusion that is corroborated by the results presented, i.e., sulfate is the dominant **non-refractory** Antarctic aerosol **by number**.

**Specific comments:**

P 2, l 8: How an aerosol affects the radiative balance *and* cloud microphysics.

P 5, l 10: In the responses, the authors state the extra vaporizing laser that measures BC was not used during 2ODIAC. This needs to be clarified here in the methods.

P 5, l 15: Please be more specific here, since composition can include both the AMS and filters. Provide a few details of what will be "future work". Publications on the filter samples? That way it is clear to the reader what will and will not be presented.

P 5, l 17: I am guessing this temperature is Celsius, but be sure to provide the units.

Methods section: Be sure to provide the time resolution for each of the instruments. I see they are provided in the R&D for some measurements, but not all. Additionally, in the beginning of the R&D, the authors express that the data are averaged every 2-minutes (and some are averaged hourly as shown in the figures). What are the standard deviations for these? The authors might want to consider showing these in the figures, or at least the SI.

P 6 , l 18: With a Pearson's of 0.32, this statement still seems like an over interpretation of the results. Perhaps tone down this deduction, or at least explain that periods with high WS typically had lower aerosol number but higher sulfate mass (at least this is what it looks like in Fig. 1). It would be good to provide the average particle number and sulfate mass concentrations during high WS and low WS time periods at some threshold; that will demonstrate what the authors are trying to conclude. Also, with the issue of blowing snow that the authors bring up, shouldn't the time periods with WS > 8 m s$^{-1}$ be eliminated (based on their statement on P 22, l 31), thus that would change the correlation coefficient? In general with linking the WS to particle number and sulfate mass, the authors should take care in clearly stating if the data were or were not eliminated and why. As is, they state data > 8 m s$^{-1}$ were excluded later in the paper but then also discuss the trends during these high WS time periods earlier on.

Section 3.1: This section contains information that seems out of order. First, the first paragraph basically concludes that sulfate is omnipresent and the main contributor to Antarctic aerosol, thus should be placed in the following section (3.2). Second, the authors do a nice job of highlighting the climate relevance of external mixtures, but the remaining paragraphs seem more like a "broader implications" section that should be placed before the conclusions. If this section were broken up and placed in the relevant locations, the order and flow would make more sense.

P 8, l 14-15: Without calculating the radiative forcing for the current work, linking to previous estimates does not connect. How does this work support the RF estimate from previous work, do the authors instead mean this work supports the fact that sulfate is a major contributor to Antarctic aerosol based on previous work, which could then impact the RF by -1 W m$^{-2}$?

P 8, l 17: The methods state that the system had a 50% transmission efficiency at 5 μm and a 0% at > 9 μm. The aerodynamic lens permits submicron particles, so where did the information in the methods come from?

Header to section 3.2: By mass? By number? By both? Please be more specific.

P 10, l 26: Define SEMS.

P 10, l 26-27: According to the Aerodyne website, the SP-AMS measures particles from 40 nm to 1 μm, thus this statement is incorrect. The authors have the 40 – 250 nm particles to work with to elucidate the composition of these transitional aerosol.

P 11, l 4-6: This statement contradicts the main conclusion that sulfate is the dominant aerosol. What percentage of the time during the studies were these conditions met? Perhaps it was for a short time period, thus not influencing the overall composition that much. On a similar note, the authors discuss the solar irradiance measurements several times; it would help to show a solar irradiance trace in Fig. 1.

P 11, l 8: This is in the appendix, not the SI. Along these lines, what is the purpose of having an appendix and a supporting information? It would be simpler if all the appendix information were placed in the supporting information, i.e., all in one referenced location.

P 12, l 23-24: Higher relative to what, to the Austral spring as presented in the current study? The authors have the data during an Austral summer to support this claim (i.e., the 2014 field season), why not validate this statement in the context of the reconstructed MSA concentrations? Although, the concentrations are low relative to previous studies…perhaps this is solely location-dependent and not on the season as the authors state. Some clarification could be used here.

P 13, l 5-10: Even though this PMF method is novel, the reasoning behind classifying Factor 3 as sulfate, Factor 2 as aged biogenic, and Factor 3 as biogenic/MSA must be based on something. How exactly were these classified? Factors 2 and 1 are very close, what is the distinguishing factor? Are there any previous studies that had factors that at least support how the current ones were classified? Or were these based on known substances measured in a laboratory setting? More explanation and clarification on how these types were classified is needed.

P 13, l 5: Why is the most abundant the last Factor?

P 13, l 12: **non-refractory** mass. Please take care in clarifying here and throughout the manuscript.

P 13, l25: Define "$Q/Q_{exp}$".

Fig. 6: Are these based on an average of single-particles or a bulk sample? How many spectra contributed to each factor? Even though the authors did add more information on PMF from the first draft, more information is still needed.

Fig. A1: Label the axes similar to the previous figures, i.e., with Austral spring and summer.

---

## Author Response (AR2)

The authors thank reviewer 1 for their further comments. It seems the reviewer is uncertain about how the AMS was operated, and how the processing of the data was done. We will address that below, but are unsure as to why the reviewer thought single particle data was used. A word search for single particle in the manuscript returned no results. Below, the reviewer comments are given in red, with responses in **black.**

 **General comments:**
The main concern I have is regarding the AMS results, namely determining the size of the sulfate aerosol. I am no expert when it comes to AMS, but doesn't the AMS only measure single particles when there is sufficient mass in each particle? Jayne et al. (2000) report on the AMS, "...the current sensitivity of this AMS for single particle counting with 100% efficiency is $\sim 2 \times 10^{-14}$ gm ($\sim 300$ nm for a pure component particle) and the sensitivity for the signal averaging mode is $\sim 0.25$ µg m$^{-3}$ for several minutes of signal integration." Considering the authors report sulfate aerosol sizes of < 300 nm ($\sim 250$ nm), wouldn't the AMS need to integrate and average > 1 particle? Further, according to Onashe et al. (2012), "The SP-AMS, like other AMS instruments, provides average mass spectrometric measurements of the ensemble of sampled aerosols rather than measurements on a particle-by particle basis." In light of this information, the authors should be clear on how the size of specifically sulfate aerosol was delineated from the measurements. Was sufficient mass present in each particle to actually measure the size *and* composition of single particles? If a bulk sample was measured, how can the authors be sure the sizes measured were purely sulfate particles? Further, the authors focus on differentiating internal versus external mixtures, which is indeed important. Although I am happy to see this distinguished, because the methods regarding how single particles were measured are not described in the level of detailed needed, it is not clear how the authors can differentiate external mixtures. If AMS integration and averaging for > 1 particle was done, how can external versus internal mixing state be determined? Perhaps this concern originates from my naïve knowledge of the AMS, but the authors should at least clarify how they came up with the size and composition of single particles to delineate sulfate aerosols.
T. B. Onasch , A. Trimborn , E. C. Fortner , J. T. Jayne , G. L. Kok , L. R. Williams , P. Davidovits & D. R. Worsnop (2012) Soot Particle Aerosol Mass Spectrometer: Development, Validation, and Initial Application, Aerosol Science and Technology, 46:7, 804-817, DOI: 10.1080/02786826.2012.663948.
J. T. Jayne , D. C. Leard , X. Zhang , P. Davidovits , K. A. Smith , C. E. Kolb & D. R. Worsnop (2000) Development of an Aerosol Mass Spectrometer for Size and Composition Analysis of Submicron Particles, Aerosol Science and Technology, 33:1-2, 49-70, DOI: 10.1080/027868200410840.
It doesn't appear that the results support the main conclusion that sulfate is the dominant Antarctic aerosol, at least, not at this level of breadth. Figure A1 shows the percent of sulfate to total AMS mass, but the percentage is < 40% most of the time, and doesn't seem to equate to the average value the authors present in the text on P 7 l 8-9, "Over both field seasons, sulfate generally makes up more than 50% of the total mass of the traditionally reported non-refractory species (organics, sulfate, nitrate, and ammonium)." Further, the authors affirm that this value is even less when considering refractory Na and Cl (5-30%). This also does not include other non-refractory species like mineral dust. By number, I can see how this conclusion holds true (i.e., based on Fig. 4). A few things here: (1) Where did the Na and Cl come from? How were these measured? (2) Based on the concern regarding how sulfate number was derived, how can this hold true? (3) If sulfate particles were indeed measured on a particle-by-particle basis, thus size could be measured by AMS, the authors should provide a more explicit conclusion that is corroborated by the results presented, i.e., sulfate is the dominant *non-refractory* Antarctic aerosol *by number*.

The AMS measurement principle is generally based on averaged particle data either bulk non-size resolved mass concentrations of various species or size resolved composition using what we call particle time of flight (PToF) mode. This mode is how we determine the size distribution of sulfate for this manuscript, and it relies on averaged data (__not__ single particle data). The various modes of the time-of-flight based AMS instruments is described in DeCarlo et al. (2006). We realize these details are not as well understood by non-AMS users and have added that clarification to the text.

*"The PToF mode in the AMS is a mode of operation in which the size-resolved aerosol composition is determined by averaging many particles during the PToF operation time (2 minutes in this study). The PToF mode does not represent single particles but rather the size distribution of particles in which ions, such as sulfate ions, are present in."*

We also note that the current data acquisition card used on our SP-AMS is capable of true "single particle" measurements but that data is not presented here, as it is a relatively new capability, and we are still analyzing that data.

Further, the reviewer should note that the Jayne et al. (2000) paper was using a quadrupole MS whereas the SP-AMS here was using the HR-ToF MS. The HR-ToF is much more sensitive than the quad MS with one-minute detection limits <0.04 µg m$^{-3}$ (DeCarlo et al., 2006) though this is somewhat variable on an instrument-by-instrument and deployment-to-deployment basis. The detection limit for the Antarctica deployments of this instrument was calculated at 10 and 12 ng m$^{-3}$ for the 2014 and 2015 campaigns, respectively. The detection limit values have been added to the text.

*"The detection limit of sulfate in the AMS was calculated to be 10 and 12 ng m-3 for the 2014 and 2015 deployments, respectively (Jayne et al., 2000)."*

With that background and explanation we can respond to some of the reviewers other comments:

Was sufficient mass present in each particle to actually measure the size *and* composition of single particles? If a bulk sample was measured, how can the authors be sure the sizes measured were purely sulfate particles?

As mentioned above, we do not present single particle data here, nor do we mean to imply it. This has again been clarified in the text (see below). In regards to the reviewer's second point: the conclusion that the particles measured in the AMS at the 250nm distribution are primarily sulfate is based off of several pieces of information:
1) comparison to the sizing instruments (Fig. 3, note that Fig. 3 is averaged over long integration times to smooth out curves and that there are transmission and collection efficiency differences between the instruments),
2) that no other species besides sulfate exhibit a well-defined peak at 250nm in the AMS PToF, and
3) Rapid changes in meteorological conditions lead to abrupt shifts in other chemical species, but not aerosol sulfate. If particles were internally mixed then all particles would be impacted similarly. This has been clarified in section 3.1 of the text:

"Taken together, the fact that neither total aerosol counts nor aerosol sulfate fall below a minimum value suggests that the background Antarctic aerosol number population may be primarily composed of sulfate. The relatively constant mass concentration of sulfate, independent of wind speed, wind direction and air mass origin, would indicate that this background aerosol is relatively temporally and geographically invariant. Other chemical species showed strong dependence on meteorological

conditions in contrast to sulfate.  If the background aerosol is primarily composed of sulfate species, then any enhancements in total aerosol counts would have to be a separate aerosol number population. An independent, externally mixed sulfate mode would therefore be expected in the AMS particle time-of-flight (PToF) mode.  A large-scale, externally mixed sulfate mode would be expected to maintain a consistent size, during variations in wind speed and direction. In both 2014 and 2015, the (inorganic) sulfate species in the AMS exhibited such a wind-independent mass distribution. Figure 2 shows the sulfate vacuum aerodynamic diameter distributions from the AMS for the 2014 field season as a function of wind speed and direction. Regardless of where an air mass originated from, the sulfate aerosol as measured by the AMS exhibited a well-distributed mode centered at a vacuum aerodynamic diameter of approximately 250nm. This result is consistent with previous ocean based size measurements of sub-Antarctic MSA containing aerosol (250 and 370nm, from Zorn et al., 2008 and Schmale et al., 2013, respectively) as well as off-line filter integrated measurements in coastal Antarctica (200-350nm aerodynamic diameter impactor stage, Jourdain and Legrand, 2001). None of the other species measured in the AMS showed a well-defined size mode. "

Further, the authors focus on differentiating internal versus external mixtures, which is indeed important. Although I am happy to see this distinguished, because the methods regarding how single particles were measured are not described in the level of detailed needed, it is not clear how the authors can differentiate external mixtures. If AMS integration and averaging for > 1 particle was done, how can external versus internal mixing state be determined? Perhaps this concern originates from my naïve knowledge of the AMS, but the authors should at least clarify how they came up with the size and composition of single particles to delineate sulfate aerosols.

This has been addressed in the above response.  Note again that the data presented is not based on single particle data, but rather PTOF or averaged data.

It doesn't appear that the results support the main conclusion that sulfate is the dominant Antarctic aerosol, at least, not at this level of breadth. Figure A1 shows the percent of sulfate to total AMS mass, but the percentage is < 40% most of the time, and doesn't seem to equate to the average value the authors present in the text on P 7 l 8-9, "Over both field seasons, sulfate generally makes up more than 50% of the total mass of the traditionally reported non-refractory species (organics, sulfate, nitrate, and ammonium)." Further, the authors affirm that this value is even less when considering refractory Na and Cl (5-30%).

While we agree and have shown that sulfate is not the dominant mass contribution to Antarctic aerosol especially when including refractory Na and Cl signals (to be discussed in a forthcoming manuscript).  However, due to the smaller size in comparison to NaCl particles, sulfate particles are predominant contributors to the aerosol number population during our measurement campaigns.  For clarity we have retitled section 3.2:

Sulfate as a Predominant Component of the Background Antarctic Aerosol Number Population

This also does not include other non-refractory species like mineral dust.

The reviewer is correct in that this study does not include mineral dust. However, both the literature and our own measurements suggest that mineral dust loadings were minor to non-existent during the 2ODIAC campaign, and more importantly would not be large contributions to the aerosol number population given their larger size.

Since submission we have obtained additional filter based results for Mg and Ca from the filters (to be discussed in an upcoming manuscript) suggest that both Mg:Cl and Ca:Cl are indicative of sea-spray derived aerosols, not crustal material derived aerosols. This is not surprising since our wind fetch was purposely selected to avoid some of the potential local dust sources (e.g Ross Island and the Dry Valleys).

Additionally, the 2 closest studies in the literature both suggest extremely low dust loadings over Antarctica: a South Pole study suggested mineral dust derived aerosol Al and Fe concentrations are <1 ng m$^{-3}$ and a coastal site (Georg von Neumayer) showed Mn concentrations at 0-30 ng SCM$^{-1}$. These two studies appear to also be in agreement with snow and ice-pack studies that suggest extremely low (1ppb) loadings of crustal material in Antarctica.

Cunningham, W. and Zoller, W.: The Chemical Composition of Remote Area aerosols, J. Aerosol Sci., 12(4), 367-384, 1981.

Wagenbach, D. et al., Coastal Antarctic aerosol: the seasonal pattern of its chemical composition and radionuclide content, Tellus B., 40B(5), 426-436, 1988.

Delmonte, B., J. R. Petit, K. K. Andersen, I. Basile-Doelsch, V. Maggi, and V. Lipenkov (2004b), Dust size evidence for opposite regional atmospheric circulation changes over East Antarctica during the last climatic transition, Clim. Dyn., 23, 427–438.

Jung-Ho Kang, Heejin Hwang, Sang Bum Hong, Soon Do Hur, Particle Size Distribution Analysis of Mineral Dust in Polar Snow Using a Coulter Counter, Ocean and Polar Research, 2014, 36, 4, 319

Lambert, F.; Bigler, M.; Steffensen, J. P.; Hutterli, M.; Fischer, H.. 2012 Centennial mineral dust variability in high-resolution ice core data from Dome C, Antarctica. Climate of the Past, 8 (2). 609-623. 10.5194/cp-8-609-2012

By number, I can see how this conclusion holds true (i.e., based on Fig. 4). A few things here: (1) Where did the Na and Cl come from? How were these measured?

The literature is clear that coastal Antarctic aerosols are heavily influenced by the ocean and thus NaCl is *primarily* of oceanic origin and likely sea spray. Some Na and Cl is always measurable in the AMS but full quantification of it is difficult and has been the subject of focused laboratory studies to be discussed in an upcoming manuscript. We now provide sulfate contribution to non-refractory species and refractory Na and Cl plus non-refractory species. This gives the reader all of the values in proper context for comparison to the existing literature.

(2) Based on the concern regarding how sulfate number was derived, how can this hold true?

From the above discussion, and points below, the reviewer's point is taken that the manuscript should be more explicit in its statement. We have included the following text in section 3.2 to clarify the manner in which we calculated the sulfate number value:

*"This process assumes sphericity of particles takes the mass distribution measured in vacuum aerodynamic diameter space, and using the density value transforms it to volume vs volume equivalent diameter for that sulfate mode, which can then be converted into a number distribution and integrated to find the total number of particles from the measured sulfate mass distribution. "*

(3) If sulfate particles were indeed measured on a particle-by-particle basis, thus size could be measured by AMS, the authors should provide a more explicit conclusion that is corroborated by the results presented, i.e., sulfate is the dominant **non-refractory** Antarctic aerosol **by number**.

Again, the AMS is not reporting single particle measurement in the context of this paper, but rather an averaged mass distribution as a function of vacuum aerodynamic diameter. This is converted into a

number population as described above. The fraction of the particle population is not limited to non-refractory particles, since the EPC used in the denominator does not discriminate between refractory and non-refractory particles. So we agree that sulfate is the dominant Antarctic aerosol by number, we do not need to add "non-refractory" to the statement as a qualifier.

**Specific comments:**
P 2, l 8: How an aerosol affects the radiative balance *and* cloud microphysics.
The reviewer's suggestion has been added to the text.
New text: "…on the global radiative balance and cloud microphysics."

P 5, l 10: In the responses, the authors state the extra vaporizing laser that measures BC was not used during 2ODIAC. This needs to be clarified here in the methods.
The text now reads:
"…The soot vaporizing laser was not used in this study."

P 5, l 15: Please be more specific here, since composition can include both the AMS and filters. Provide a few details of what will be "future work". Publications on the filter samples? That way it is clear to the reader what will and will not be presented.
The text now reads:
"…other parts of the instrumentation suite, including the particle filters and snow samples…"

P 5, l 17: I am guessing this temperature is Celsius, but be sure to provide the units.
"C" has been added to the text, the text now reads:
"…3°C and 3% accuracy, respectively…"

Methods section: Be sure to provide the time resolution for each of the instruments. I see they are provided in the R&D for some measurements, but not all. Additionally, in the beginning of the R&D, the authors express that the data are averaged every 2-minutes (and some are averaged hourly as shown in the figures). What are the standard deviations for these? The authors might want to consider showing these in the figures, or at least the SI.
The time resolution for each instrument has been added in the instrumentation section. The hourly interval is only included in Fig.1B to aid in readability in the figure and is not included in any analysis. The standard deviation for the 2-minute average of Fig 1A and 1B has been included in the figure caption. This has been calculated by averaging the standard deviation for each 2-minute averaged period of 1-second resolution data since the reviewer's question seems to deal with the variability within each 2-minute averaged period and not the record as a whole.

P 6 , l 18: With a Pearson's of 0.32, this statement still seems like an over interpretation of the results. Perhaps tone down this deduction, or at least explain that periods with high WS typically had lower aerosol number but higher sulfate mass (at least this is what it looks like in Fig. 1). It would be good to provide the average particle number and sulfate mass concentrations during high WS and low WS time periods at some threshold; that will demonstrate what the authors are trying to conclude. Also, with the issue of blowing snow that the authors bring up, shouldn't the time periods with WS > 8 m s$^{-1}$ be eliminated (based on their statement on P 22, l 31), thus that would change the correlation coefficient? In general with linking the WS to particle number and sulfate mass, the authors should take care in clearly stating if the data were or were not eliminated and why. As is, they state data > 8 m s$^{-1}$ were excluded later in the paper but then also discuss the trends during these high WS time periods earlier on.

The reviewer is correct in pointing out that high wind speed conclusions are not germane to this manuscript. The correlation between wind speed and particle number concentrations is not as relevant to this manuscript as it is to the forthcoming blowing snow manuscript so this statement has been removed from the text. The main focus of this section of the manuscript is to draw attention to the minimum particle number concentration so the average loadings as a function of wind speed will also be relegated to a future publication.

With regards to the reviewer's comments on other mentions of high wind speeds, these have been clarified in the text with the statement:

"…since high wind speeds and particle counts are strongly correlated, to be discussed in a forthcoming manuscript"

Section 3.1: This section contains information that seems out of order. First, the first paragraph basically concludes that sulfate is omnipresent and the main contributor to Antarctic aerosol, thus should be placed in the following section (3.2). Second, the authors do a nice job of highlighting the climate relevance of external mixtures, but the remaining paragraphs seem more like a "broader implications" section that should be placed before the conclusions. If this section were broken up and placed in the relevant locations, the order and flow would make more sense.

The authors feel that the first paragraph sets the hypothesis that sulfate is a key background aerosol, by stating the observation that neither sulfate or particle counts every drop to 0. The remainder of the paragraph examines this hypothesis in more detail offering additional observations to strengthen this hypothesis.

P 8, l 14-15: Without calculating the radiative forcing for the current work, linking to previous estimates does not connect. How does this work support the RF estimate from previous work, do the authors instead mean this work supports the fact that sulfate is a major contributor to Antarctic aerosol based on previous work, which could then impact the RF by -1 W m$^{-2}$?

We feel that this inclusion is important to provide additional importance for the observation of an external mixture to a broader audience. While we do not explicity calculate radiative forcing, we are are mentioning the work of others (e.g. Myhre et al.) who, based on our work, correctly make this assumption.

P 8, l 17: The methods state that the system had a 50% transmission efficiency at 5 μm and a 0% at > 9 μm. The aerodynamic lens permits submicron particles, so where did the information in the methods come from?

The transmission efficiency in the methods is of the aerosol inlet to the inlet of the AMS, not the AMS's transmission efficiency. P5 L10-14:

"The inlet for the aerosol sampling line was covered and heated to prevent sampling of wind blown snow and to prevent riming, respectively. At the flow conditions and geometry of the sampling inlet, transmission of <1μm particles to the AMS was >95%. Overall, the system had a 50% transmission efficiency at 5μm and 0% transmission efficiency of particles >9μm. All transmission efficiency values are as calculated by a particle loss calculator using the specific geometry of the setup (von der Weiden et al., 2009)."

P8 L17 explicitly states "aerodynamic lens of the AMS" but the text on P5 has been changed to:
"…Overall, the aerosol inlet system had a 50% transmission efficiency at 5μm and 0% transmission efficiency of particles >9μm. …"

Header to section 3.2: By mass? By number? By both? Please be more specific.
Section 3.2 has been renamed to: Sulfate as a Predominant Component of the Background Aerosol Number Population. See below.

P 10, l 26: Define SEMS.
SEMS has already been previously defined in Section 2.2.

P 10, l 26-27: According to the Aerodyne website, the SP-AMS measures particles from 40 nm to 1 µm, thus this statement is incorrect. The authors have the 40 – 250 nm particles to work with to elucidate the composition of these transitional aerosol.
The reviewer is citing the aerodynamic lens range transmission but what matters to the statement on P10,L26-27 is the mass loading in PToF mode. In PToF mode, the mass loading of particles is greatly reduced as opposed to "open" mode. In the transitional period, the total aerosol number is low (200-400 #/cc) which, even at low detection limits in the AMS, translates to extremely small mass loadings at particle diameters of 40-150nm.  Note that the AMS is sensitive to mass not number, so a 50 nm particle has 125 times less mass than a 250 nm particle.  The fact that there is a measurable sulfate distribution on the lower end of the distribution curve is testament to how sensitive the AMS is given how little sulfate there is in Antarctic aerosol compared to other environments. The statement has been changed to clarify this point:

"Unfortunately, given the instrumentation deployed, it is impossible to determine the composition of these very small particles (e.g. less than 100 nm) due in part to the low mass loadings inherent in Antarctica and the effects that low mass loadings have on AMS PToF data."

P 11, l 4-6: This statement contradicts the main conclusion that sulfate is the dominant aerosol. What percentage of the time during the studies were these conditions met? Perhaps it was for a short time period, thus not influencing the overall composition that much. On a similar note, the authors discuss the solar irradiance measurements several times; it would help to show a solar irradiance trace in Fig. 1.
As mentioned before we have retitled section 3.2 to read:
"Sulfate as a Predominant Component of the Background Antarctic Aerosol Number Population"

For the reviewer's second point, we are candid about the fact that these measurements are the first of their kind and are limited because this is the only observation of this transitional period in the record. We do not mean to suggest, nor do we believe we write, that the transitional period is important to global climate. Rather, we point to it as potential evidence of a new particle formation mechanism over Antarctica that should be closely examined in future studies.
Our solar irradiance measurements are semi-quantitative, and are not be shown in the figures due to the potential for these values to be taken as more accurate than they are. Solar irradiance in the manuscript is mentioned only to serve as a guide and a relative measurement.

P 11, l 8: This is in the appendix, not the SI. Along these lines, what is the purpose of having an appendix and a supporting information? It would be simpler if all the appendix information were placed in the supporting information, i.e., all in one referenced location.
During the review process we were asked to place this information in the appendix, which is why there is both an appendix and an SI.  Given the usefulness of the figures for interpreting the

manuscript we will continue to have the appendicies included in the main paper, and the supplementary information (less necessary for interpretation of the manuscript) separate.

P 12, l 23-24: Higher relative to what, to the Austral spring as presented in the current study? The authors have the data during an Austral summer to support this claim (i.e., the 2014 field season), why not validate this statement in the context of the reconstructed MSA concentrations? Although, the concentrations are low relative to previous studies…perhaps this is solely location-dependent and not on the season as the authors state. Some clarification could be used here.

"Higher" in this context is relative to what was measured in the early summer of 2ODIAC 2014. The sentence suggests that in mid-late summer, MSA should increase as compared to what was measured in early summer. This statement is verified in that there is virtually no MSA observed in winter-spring and it only begins to appear in early summer. We explicitly state this in the precedeing sentences of the paragraph.

The location of the measurements is mentioned because it is assumed that proximity to source is important in the interpretation of concentration measurements. From our observations and other literature (e.g. Minikin et al., 1998; Preunkert et al., 2007; Preunkert et al., 2008; Read et al., 2008;), it seems clear that MSA concentrations are both temporally and spatially dependent.

P 13, l 5-10: Even though this PMF method is novel, the reasoning behind classifying Factor 3 as sulfate, Factor 2 as aged biogenic, and Factor 3 as biogenic/MSA must be based on something. How exactly were these classified? Factors 2 and 1 are very close, what is the distinguishing factor? Are there any previous studies that had factors that at least support how the current ones were classified? Or were these based on known substances measured in a laboratory setting? More explanation and clarification on how these types were classified is needed.

The following has been added to the text to clarify why the factors were named as they were:

New text: The naming of the factors, Sulfate, Biogenic/MSA, and aged biogenic, for Factors 3, 2, and 1, respectively, was chosen based on two main aspects of the resultant PMF spectra. The sulfate factor was named due to its lack of ions containing both Carbon and Sulfur atoms, and the fact that it bears some resemblance to the mass spectra obtained by atomizing diluted sulfuric acid into the AMS. The Biogenic and Aged Biogenic factors were distinguished primarily by the contribution of the $C_xH_yS_xO_p$ ions appearing in the Biogenic factor and to a lesser extent in the Aged Biogenic factor. If this assumption is not correct then the naming schema of the last factor may need to be adjusted in the future. Again, the three factor solution was chosen due to it minimizing the $Q/Q_{exp}$ of the PMF solution without creating obviously extraneous factors (see SI Fig.S4).

Factors 1 and 2, while similar, exhibit clear differences in their mass spectra. There are large differences in the $C_xH_yS_zO_p$ ions contributions and times series of the factors. The $C_xH_yS_zO_p$ ions appear to a greater extent in Factor 2, as well as the ratios of the major sulfate peaks to each other (e.g. $SO^+$:$SO_2^+$, $SO_2^+$:$SO_3^+$, etc.) and to their hydrogenated compounds ($HSO_2^+$, $HSO_3^+$, $H_2SO_4^+$). In the AMS, these ratios are consistent and robust for specific chemical species. Factors 1 and 3 are more closely related than 1:2 or 2:3 but this is due to the fact that $m/z$ 48, 64, and 80 (major sulfate peaks) make up the majority of both factors. Even so, Factor 3 still has $C_xH_yS_zO_p$ peaks that Factor 1 does not and has differing sulfate:$H_xSO_y$ ratios from Factor 1. The R_tseries vs R_profiles output from the PMF analysis (Ulbrich et al., 2009) can be seen in SI Fig. S4.D for interested readers.

New text: Factors 1 and 2 are differentiated in both their percent contributions of the CxHySxOp ions as well as the ratios of major sulfate peaks (e.g. $SO^+$:$SO_2^+$).

PMF has almost exclusively been applied solely to organic masses from the AMS. The few studies that have elucidated MSA factors out of marine aerosol do not apply PMF to the sulfates but only the CxHySxOp ions so there are no comparable studies to compare to. The laboratory measured MSA spectra is included in SI for comparison but, as already mentioned in the text, suggests that the Biogenic/MSA factor is not solely composed of MSA due to differences in the principle ratios discussed above.

P 13, l 5: Why is the most abundant the last Factor?
The factor numbering was preserved from the PMF output and should not be taken to be indicative of any valuation on the factors themselves. The following has been added to the text to reflect this point:
New text: It should be noted that the numbering of the factors is preserved from the PMF solution and is not indicative of any valuation on the factors themselves.

P 13, l 12: *non-refractory* mass. Please take care in clarifying here and throughout the manuscript.
"Non-refractory" has been clarified in multiple places in the text.

P 13, l25: Define "$Q/Q_{exp}$".
$Q/Qexp$ has been defined in the text at the beginning of S3.3.1.
New Text: PMF analysis uses as input any number of mass spectra as a function of time, measured in the AMS, to pick out patterns in both temporal and mass spectral phases of the data. PMF determines "factors" that contribute to the time-series and mass spectral series by minimizing a "quality of fit" parameter (the sum of the residual of a given input not fit by the algorithm scaled by estimated error in the time and mass-spectral series). The PMF algorithm is run over multiple starting points in the rotations of the matrices and limited to given numbers of factors. $Q/Q_{exp}$ is used as a metric to determine how well the error model assumed in PMF is represented by the dataset.  In general, the lowest values of $Q/Qexp$ are the best solution.  This metric is described in detail in Ulbrich et al. (2009).

Fig. 6: Are these based on an average of single-particles or a bulk sample? How many spectra contributed to each factor? Even though the authors did add more information on PMF from the first draft, more information is still needed.
As discussed above, no single particle data is used in this paper. PMF analysis uses mass spectra (of the ensemble aerosol mass, not size resolved) at given times to elucidate patterns in the MS and time series data. The input into PMF in this analysis consisted of over 35,000 individual mass spectra. PMF analyzes all of the spectra to pick out patterns that are "Factors". This has been clarified in the text:

New text: PMF analysis uses as input any number of mass spectra as a function of time, measured in the AMS, to pick out patterns in both temporal and mass spectral phases of the data…In this study, over 33,000 individual mass spectra were used as input to PMF.

Fig. A1: Label the axes similar to the previous figures, i.e., with Austral spring and summer.

The figure has been changed as per the reviewer's suggestion.